# Development and characterization of new tools for detecting poly(ADP-ribose) in vitro and in vivo

Sridevi Challa[1,2†], Keun W Ryu[1,2†‡], Amy L Whitaker[1,2,3], Jonathan C Abshier[1,2], Cristel V Camacho[1,2], W Lee Kraus[1,2*]

[1]Laboratory of Signaling and Gene Regulation, Cecil H. and Ida Green Center for Reproductive Biology Sciences, University of Texas Southwestern Medical Center, Dallas, United States; [2]Division of Basic Research, Department of Obstetrics and Gynecology, University of Texas Southwestern Medical Center, Dallas, United States; [3]Program in Genetics, Development, and Disease, Graduate School of Biomedical Sciences, University of Texas Southwestern Medical Center, Dallas, United States

**\*For correspondence:**
LEE.KRAUS@utsouthwestern.edu

[†]These authors contributed equally to this work

**Present address:** [‡]Cancer Biology & Genetics Program, Memorial Sloan Kettering Cancer Center, New York, United States

**Abstract** ADP-ribosylation (ADPRylation) is a reversible post-translation modification resulting in the covalent attachment of ADP-ribose (ADPR) moieties on substrate proteins. Naturally occurring protein motifs and domains, including WWEs, PBZs, and macrodomains, act as 'readers' for protein-linked ADPR. Although recombinant, antibody-like ADPR detection reagents containing these readers have facilitated the detection of ADPR, they are limited in their ability to capture the dynamic nature of ADPRylation. Herein, we describe and characterize a set of poly(ADP-ribose) (PAR) Trackers (PAR-Ts)—optimized dimerization-dependent or split-protein reassembly PAR sensors in which a naturally occurring PAR binding domain, WWE, was fused to both halves of dimerization-dependent GFP (ddGFP) or split Nano Luciferase (NanoLuc), respectively. We demonstrate that these new tools allow the detection and quantification of PAR levels in extracts, living cells, and living tissues with greater sensitivity, as well as temporal and spatial precision. Importantly, these sensors detect changes in cellular ADPR levels in response to physiological cues (e.g., hormone-dependent induction of adipogenesis without DNA damage), as well as xenograft tumor tissues in living mice. Our results indicate that PAR Trackers have broad utility for detecting ADPR in many different experimental and biological systems.

## Editor's evaluation

Challa and Ryu et al., systematically evaluated various combinations of ADP-ribose-binding modules to make sensors detecting poly(ADP-ribose). A series of GFP- and luciferase-based sensors has been created by the authors who demonstrated their applications in vitro, in living cells, and, for the first time, in animals. Despite these sensors still needing to be improved upon their dynamic ranges and signal/background ratios, these tools open the possibility to detect ADP-ribosylation in many experimental and biological systems in vitro and in vivo.

## Introduction

ADP-ribosylation (ADPRylation) is a regulatory post-translational modification (PTM) of proteins that results in the reversible attachment of ADP-ribose (ADPR) units on substrate proteins (*Gupte et al., 2017*; *Lüscher et al., 2018*). Members of the PARP family of ADP-ribosyltransferases (ARTs) (*Amé et al., 2004*; *Vyas et al., 2013*) function as 'writers' to catalyze the transfer of ADPR moieties from

oxidized β-nicotinamide adenine dinucleotide (NAD⁺) to a variety of amino acids (Asp, Glu, Ser, Arg, and Lys) in substrate proteins (*Gupte et al., 2017*; *Lüscher et al., 2018*). Mono(ADP-ribosyl) transferases (MARTs) add a single ADPR moiety to their substrates through a process called mono(ADP-ribosyl)ation (MARylation) (*Challa et al., 2021*), whereas poly(ADP-ribosyl) transferases (PARPs) add branched or linear chains of multiple ADPR moieties through a process called poly(ADP-ribosyl)ation (PARylation) (*Gupte et al., 2017*; *Lüscher et al., 2018*). Site-specific ADPRylation of substrate proteins by PARP enzymes can have important functional consequences, including alteration of the biochemical or biophysical properties of the substrate protein or creation of new binding sites for ADPR binding domains (ARBDs) that drive protein-protein interactions (*Gibson and Kraus, 2012*; *Gupte et al., 2017*). As such, ADPRylation can control a wide variety of cellular and biological processes, including DNA repair, DNA replication, gene expression, and RNA biology, as well as inflammatory responses and cell differentiation (*Challa et al., 2021*; *Gibson and Kraus, 2012*; *Gupte et al., 2017*; *Lüscher et al., 2018*).

The various forms of ADPR are recognized and bound by an assortment of protein domains and motifs that are found in a variety of proteins with diverse functions and mediate many of the biological functions of ADPRylation (*Barkauskaite et al., 2013*; *Gibson and Kraus, 2012*; *Teloni and Altmeyer, 2016*). These ARBDs function as 'readers' of the various forms of protein-linked ADPR (MAR, PAR, branched, terminal residue, etc.). They include PAR-binding motifs (PBMs), macrodomains, PAR-binding zinc fingers (PBZs), and WWE domains (*Ahel et al., 2008*; *Feijs et al., 2013*; *Gagné et al., 2008*; *Karras et al., 2005*; *Pleschke et al., 2000*; *Rack et al., 2016*; *Wang et al., 2012*). Macrodomains recognize free ADPR, as well as the terminal ADPR moieties in PAR, allowing them to bind to both MAR and PAR (*Karras et al., 2005*; *Timinszky et al., 2009*). PBZ domains recognize branched forms of PAR (*Chen et al., 2018*). WWE domains recognize the iso-ADPR linkages joining ADPR monomers, restricting their binding to PAR (*Kang et al., 2011*; *Wang et al., 2012*; *Zhang et al., 2011*). In addition to the 'writers' and 'readers,' 'eraser' enzymes, including PAR glycohydrolase (PARG) and ADP-ribosylhydrolase 3 (ARH3), recognize specific ADPR modifications through the ARBDs and catalyze PAR chain degradation through endo- and exoglycocidic activities. Their activities leave the terminal ADPR moiety attached to the acceptor amino acid residue of the substrate (*Barkauskaite et al., 2013*; *Niere et al., 2012*; *Oka et al., 2006*; *Slade et al., 2011*).

Although recent developments in mass spectrometry-based identification of ADPR-modified amino acids have enhanced the study of specific ADP-ribosylation events on target proteins (*Daniels et al., 2015*), the lack of a complete set of immunological tools that recognize the diverse forms of ADPR has hampered progress in studying ADPRylation. Anti-ADPR polyclonal antibodies have been reported, but the specificity and utility of these antibodies have not been assessed broadly (*Bredehorst et al., 1978*; *Kanai et al., 1974*; *Kanai et al., 1978*; *Meyer and Hilz, 1986*; *Sakura et al., 1978*). Instead, the PARP field has relied on the anti-PAR monoclonal antibody 10H, which binds to PAR chains longer than ten ADPR units (*Kawamitsu et al., 1984*). Although useful, this antibody has left the field blind to mono- and oligo(ADP-ribosyl)ation.

The recent development of recombinant site-specific and broad-specificity antibodies to ADPR has been a major advance (*Bonfiglio et al., 2020*). We recently described the generation and characterization of a set of recombinant antibody-like ADP-ribose binding proteins, in which natural ARBDs have been functionalized with the Fc region of rabbit immunoglobulin (*Gibson et al., 2017*). Collectively, the ADPR detection tools described here are useful for cellular and biochemical assays, but they are not useful for exploring the dynamics of ADPRylation in cells and in animals. In fact, ADPRylation is a rapid process that can occur within minutes and can be removed by various 'erasers' including ARH3 and PARG (*Barkauskaite et al., 2013*; *Gupte et al., 2017*). Therefore, developing tools to measure ADPR dynamics in cells and in vivo is critical for better understanding the various biological processes mediated by ADPR.

To overcome this limitation, several split-protein reassembly approaches have been applied to PAR detection. With this approach, nonfunctional fragments of a split-fluorescent protein or luciferase are induced to reassemble through the direct interaction of fused ARBDs (*Furman et al., 2011*; *Krastev et al., 2018*; *Lee et al., 2021*; *Serebrovskaya et al., 2020*). These include the PBZ modules of aprataxin PNK-like factor (APLF) (*Ahel et al., 2008*) attached to each half of split firefly luciferase (split-Fluc) (*Furman et al., 2011*), PBZ modules with split Venus GFP (*Krastev et al., 2018*), and WWE domains with Turquoise and Venus, allowing for Förster resonance energy transfer (FRET)

(*Serebrovskaya et al., 2020*). However, these tools have some limitations: (1) they can only detect PAR accumulation in vitro (*Furman et al., 2011*), (2) they can only detect PAR accumulation on specific target proteins (*Krastev et al., 2018*), or (3) they have modest dynamic ranges (*Serebrovskaya et al., 2020*). Moreover, none of these sensors has been shown to be capable of detecting PAR accumulation in vivo.

In the work described herein, we developed a set of PAR Trackers (PAR-Ts)—optimized dimerization-dependent and split-protein reassembly PAR detection tools that have broad utility for both in vitro and in vivo studies. The PAR-Ts contain a WWE fused to both parts of dimerization-dependent GFP (ddGFP) (*Alford et al., 2012*) or split Nano luciferase (NanoLuc) (*Wang et al., 2020*) with LSSmOrange. The ddGFP version (PAR-T ddGFP) allows for real-time assessment of dynamic PAR production in vitro and in living cells, while the split NanoLuc version (PAR-T NanoLuc) allows detection of PAR production in tissues in living mammals.

## Results

The use of functionalized ARBDs to detect ADP-ribosylation has been a useful approach (*Forst et al., 2013*; *Gibson et al., 2017*; *Timinszky et al., 2009*). In this study, we further developed our previous ADPR detection reagents (*Gibson et al., 2017*) to expand their utility to in vivo applications as ADPR sensors. To achieve this, we used a systematic approach including in vitro characterization of the sensors and validation of their utility in vivo.

**Table 1.** Nomenclature, composition, and activity of the various PAR-T sensors used in this study.
A summary of the PAR-Tracker sensors generated in this study. The activity of these sensors in vitro and in cells, and the corresponding figures in which the activities are described, are indicated. The activities are described as low (+), medium (++), and high (+++, ++++). N.D. not determined.

| Nomenclature used in the manuscript | ARBDs used with fluorescent or luminescent protein fragments | | Activity in vitro | Activity in cells | Figures |
|---|---|---|---|---|---|
| **Fluorescent sensors** | **ddGFP-A** | **ddGFP-B** | | | |
| PAR-T ddGFP | WWE from RNF146 | WWE from RNF146 | +++ | ++++ | *Figure 1—figure supplement 1D, E* |
| | Macrodomain from AF1521 | Macrodomain from AF1521 | ++ | N.D. | *Figure 1—figure supplement 1D, E* |
| | PBZ from APLF | PBZ from APLF | + | N.D. | *Figure 1—figure supplement 1D, E* |
| | Macrodomain from MH2A.1 | Macrodomain from MH2A.1 | + | N.D. | *Figure 1—figure supplement 1D, E* |
| | WWE from RNF146 | Macrodomain from AF1521 | +++ | N.D. | *Figure 1—figure supplement 1D, E* |
| | Macrodomain from AF1521 | WWE from RNF146 | ++++ | – | *Figure 1—figure supplement 1D, E* |
| **Firefly luciferase-based sensors** | **N-terminal of Firefly luciferase (FLucN)** | **C-terminal of Firefly luciferase (FLucC)** | | | |
| PAR-T fLuc | WWE from RNF146 | WWE from RNF146 | N.D. | + | *Figure 4—figure supplement 1C* |
| | Macrodomain from AF1521 | Macrodomain from AF1521 | N.D. | – | *Figure 4—figure supplement 1C* |
| | WWE from RNF146 | Macrodomain from AF1521 | N.D. | – | *Figure 4—figure supplement 1C* |
| | Macrodomain from AF1521 | WWE from RNF146 | N.D. | – | *Figure 4—figure supplement 1C* |
| **Nano luciferase-based sensors** | **N-terminal of Nano luciferase (NanoLucN)** | **C-terminal of Nano luciferase plus LssmOrange (NanoLucC)** | | | |
| PAR-T NanoLuc | WWE from RNF146 | WWE from RNF146 | N.D. | ++++ | *Figure 4C* |

## Using dimerization-dependent GFP-based sensors to detect PAR in Vitro

Dimerization-dependent GFP is a genetically encoded sensor that was initially developed to study protein interactions (*Alford et al., 2012*). In this system, a pair containing a quenched GFP (ddGFPA) and a nonfluorogenic GFP (ddGFPB) form a heterodimer with improved fluorescence (*Alford et al., 2012*). The reversible complementation of ddGFP pairs, unlike irreversible split fluorophores, is ideal to monitor dynamic signaling events, such as PARylation (*Villalobos et al., 2007*). Hence, we sought to design ddGFP-based fluorescence sensors for PAR (fluorescent PAR-Trackers or PAR-T ddGFP) (*Table 1*) to enable us to perform live-cell imaging, with a high signal-noise ratio. To achieve this, we fused various ARBDs to ddGFP-A/B and purified the recombinant proteins (*Figure 1A and B*, *Figure 1—figure supplement 1A*). We performed in vitro ADP-ribosylation assays using recombinant PARP-1 (to generate PAR) and PARP-3 (to generate MAR) (*Figure 1—figure supplement 1B and C*). We observed that of all the ARBD-ddGFP pairs tested, the WWE domain from RNF146, macrodomain from AF1521, and a combination of these two performed well in specifically recognizing PARylated PARP-1 (*Figure 1—figure supplement 1D and E*). These sensors recognized PARylated-PARP-1, but not MARylated PARP-3, or the precursors of ADPR (*Figure 1C*).

We further tested the sensitivity of these sensors and their dynamic range using in vitro PARP-1 PARylation reactions with increasing concentrations of $NAD^+$ (*Figure 1—figure supplement 1F and G*) and increasing time of reaction (*Figure 1D and E*). Similarly, we detected the degradation of PAR chains by the ADP-ribosylhydrolase ARH3 in a time (*Figure 1F and G*) and dose (*Figure 1—figure supplement 1H*) dependent manner. Further, we performed in vitro reactions by incubating the recombinant PAR-T sensors with lysates from HeLa cells treated with $H_2O_2$ to induce DNA damage and activate PARP-1, or an inhibitor of the PAR glycohydrolase PARG (i.e., PDD00017273) to increase PAR. We observed an increase in fluorescence when lysates from cells treated with either $H_2O_2$ or PARG inhibitor (PDD00017273) were used, as well as a profound increase in fluorescence when lysates from cells treated with both $H_2O_2$ and PARG inhibitor were used (*Figure 1—figure supplement 1I and J*). The $H_2O_2$- and PARG inhibitor-stimulated signals were reduced with PARP inhibitor treatment (*Figure 1—figure supplement 1I and J*). Taken together, these data suggest that the PAR-T ddGFP sensors can specifically recognize PAR, and that they exhibit a good dynamic range in vitro.

## Using PAR-T ddGFP sensors to detect PAR in live cells

Having confirmed the specificity of the PAR-T sensors, we next sought to test their utility in live-cell imaging. We expressed the PAR-T sensors in HeLa cells using doxycycline (Dox) induction and performed live-cell imaging after subjecting the cells to $H_2O_2$-mediated PARP-1 activation (*Figure 2—figure supplement 1A*). The live cell PAR-T construct also expresses mCherry with a nuclear localization signal (NLS) to illuminate the nuclei and act as a control for the expression of the constructs (*Figure 2—figure supplement 1A*). When compared to ddGFP alone, ddGFP-conjugated to WWE detected PARP-1 activation in live-cell imaging (*Figure 2A and B*). Interestingly, even though the WWE-macrodomain combination sensor was able to detect PAR in vitro, this sensor combination failed to recognize PAR in cells (*Figure 2—figure supplement 1B and C*). Hence, we used the WWE-based PAR-T sensors for the experiments from this point onwards. Using the WWE-based ddGFP PAR-T sensor, we were able to detect accumulation of PAR after $H_2O_2$-treatment in real time. Treatment with PARP inhibitor blocked this accumulation (*Figure 2C and D*; *Figure 2—video 1*).

Cancers are heterogenous tissues with spatial variation in nutrient availability and cell-extrinsic stressors (*Dagogo-Jack and Shaw, 2018*). PAR is enhanced by stressors, such as DNA damage or hypoxia, but the spatiotemporal dynamics of PAR production in cells have remained unclear due to a lack of efficient detection methods. Thus, we asked if PAR levels vary spatially in groups of cells. We performed live-cell imaging in three-dimensional (3D) cancer spheroids using MCF-7 human breast cancer cells expressing the WWE-based PAR-T-ddGFP sensor. We observed a heterogeneous distribution of PAR throughout the spheroid, which was inhibited by the PARP inhibitor, Niraparib (*Figure 3A and B*). We also performed a time course of Niraparib treatment in 3D cancer spheroids to visualize spatio-temporal changes in PAR levels over time. The results indicate that the PAR levels in cells at the core of the spheroids are relatively resistant to Niraparib treatment, since the PAR levels in these cells decrease at a lower rate compared to the PAR levels in the cells in the outer layer of the spheroid

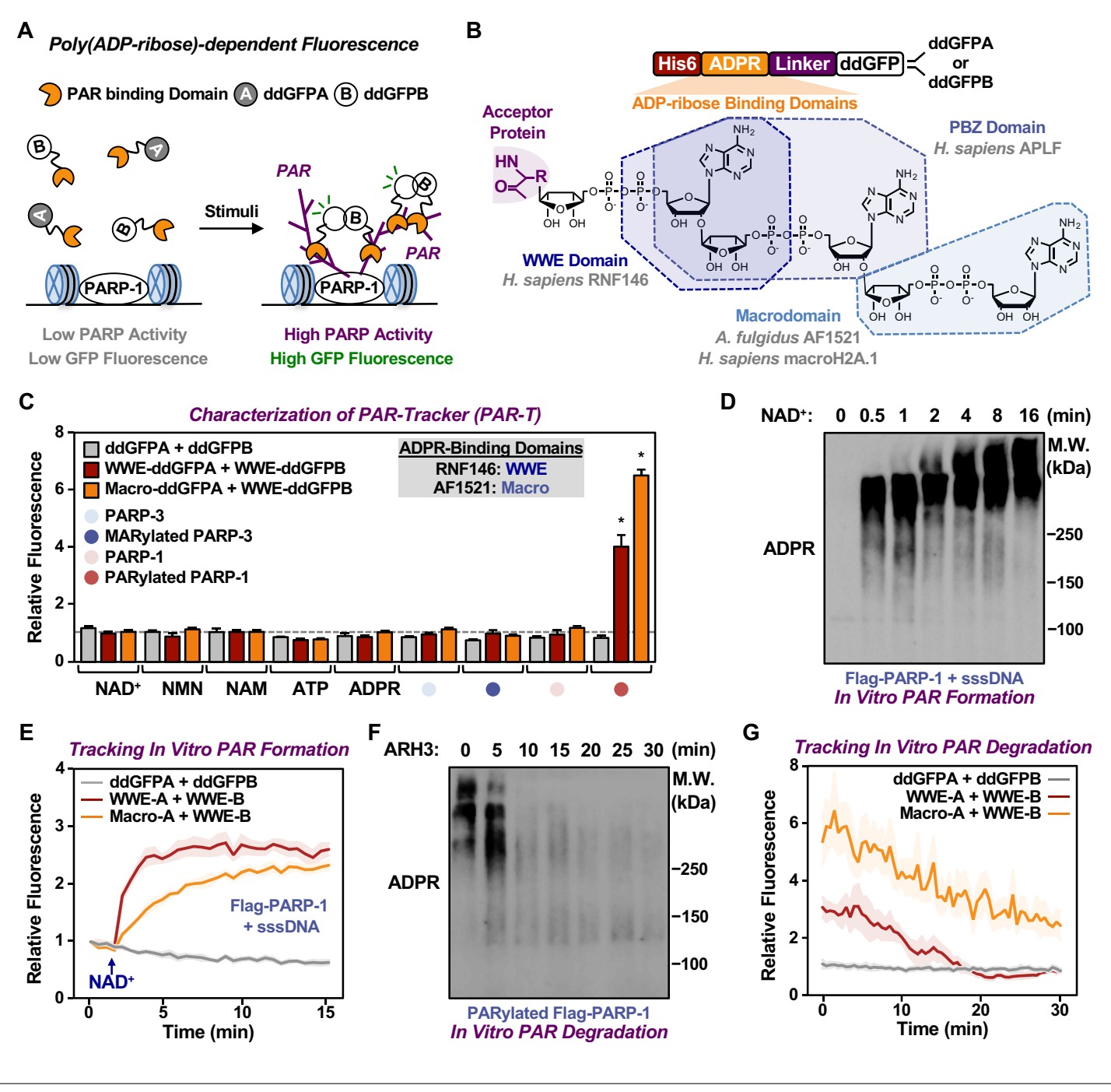

**Figure 1.** Development of ddGFP-based fluorescent sensors to measure PAR levels. (**A**) Schematic diagram of the fluorescent PAR Trackers (PAR-Ts). (**B**) Schematic diagram of the plasmid constructs used to express the ddGFP PAR-T in bacteria. Chemical structures of a PARylated amino acid, a MARylated amino acid, and the chemical moieties in ADPR that are recognized by the ADPR binding domains. The constructs contain DNA segments encoding (1) His tag (red), (2) ADP-ribose binding domain (yellow), (3) a flexible linker (purple), and (4) ddGFP proteins (white). (**C**) Verification of substrate specificity. Fluorescence measurements of in vitro ADPRylation assays containing the indicated substrates. Each bar in the graph represents the mean ± SEM of the relative fluorescence intensity (n=3, two-way ANOVA, *p<0.0001). (**D, E**) Western blot analysis (**D**) and fluorescence measurements (**E**) of the time course of in vitro PAR formation using recombinant PARP-1. Each line plot in the graph in (**E**) represents mean ± SEM of relative fluorescence intensity (n=3). (**F, G**) Western blot analysis (**F**) and fluorescence measurements (**G**) of the time course of in vitro PAR degradation using recombinant ARH3. Each line plot in the graph in (**G**) represents the mean ± SEM of relative relative fluorescence intensity (n=3).

The online version of this article includes the following figure supplement(s) for figure 1:

**Figure supplement 1.** Characterization of fluorescence-based PAR-Trackers using biochemical assays.

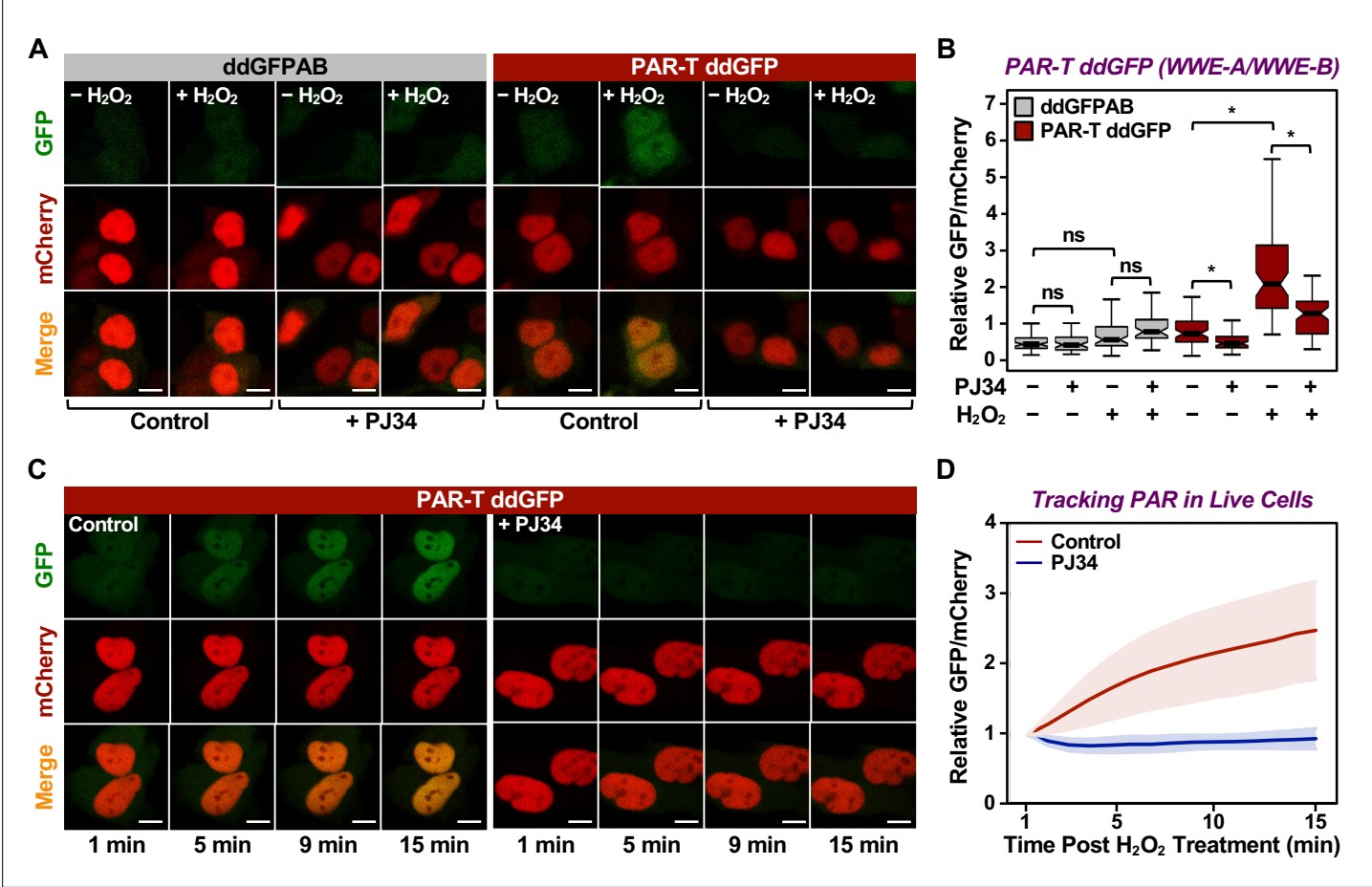

**Figure 2.** Tracking PAR accumulation in live cells using PAR-T ddGFP. (**A, B**) Live-cell imaging assay to track PAR formation in response to $H_2O_2$ in 293T cells subjected to Dox-induced PAR-T ddGFP expression. The cells were treated with 20 μM PJ34 for 2 hr prior to treatment with 1 mM $H_2O_2$ for 15 min. The scale bar is 10 μm. Each bar in the graph in (**B**) represents the mean ± SEM of the relative levels of the fluorescence intensity (n=3, one-way ANOVA, *p<0. 01; ns=not significant). (**C, D**) Live-cell tracking of PAR formation in response to $H_2O_2$. HeLa cells subjected to PAR-T ddGFP expression were treated with 20 μM PJ34 for 2 hr prior to treatment with 1 mM $H_2O_2$ and live-cell imaging. The scale bar is 10 μm. Each bar in the graph in (**D**) represents the mean ± SD of the relative levels of the fluorescence intensity (n=20 for control and n=21 for PJ34).

The online version of this article includes the following video and figure supplement(s) for figure 2:

**Figure supplement 1.** Characterization of fluorescence-based PAR-Trackers using live-cell imaging.

**Figure 2—video 1.** Tracking the levels of PAR in cells.

https://elifesciences.org/articles/72464/figures#fig2video1

(*Figure 3C and D*). These results show that the PAR-T-ddGFP sensor can be used for live-cell imaging to evaluate spatial and temporal changes in PARylation in cancer cells.

## Developing a highly sensitive split-luciferase PAR-T sensor

Since we demonstrated that WWE domain-based PAR-T sensors can specifically detect PAR in biochemical assays and living cells, we sought to develop a set of highly sensitive PAR-T sensors that can detect PAR in vivo. Fluorescent sensors are not well suited for detection in vivo due to auto-fluorescence of tissues that can cause high background signals. Instead, luminescence-based approaches are preferred for in vivo applications due to the lack of auto-luminescence in tissues (*Tung et al., 2016*). Hence, we developed a set of luminescent PAR-T sensors to detect PAR levels in vivo. We first generated sensors based on split firefly luciferase (*Maita et al., 2014*) using various combinations of the ARBDs. As before, the WWE domains consistently performed better in detecting an increase in PAR levels with PARG inhibitor treatment and a decrease in PAR levels with a PARP inhibitor treatment using either cell lysates (*Figure 4—figure supplement 1A*) or live cells (*Figure 4—figure*

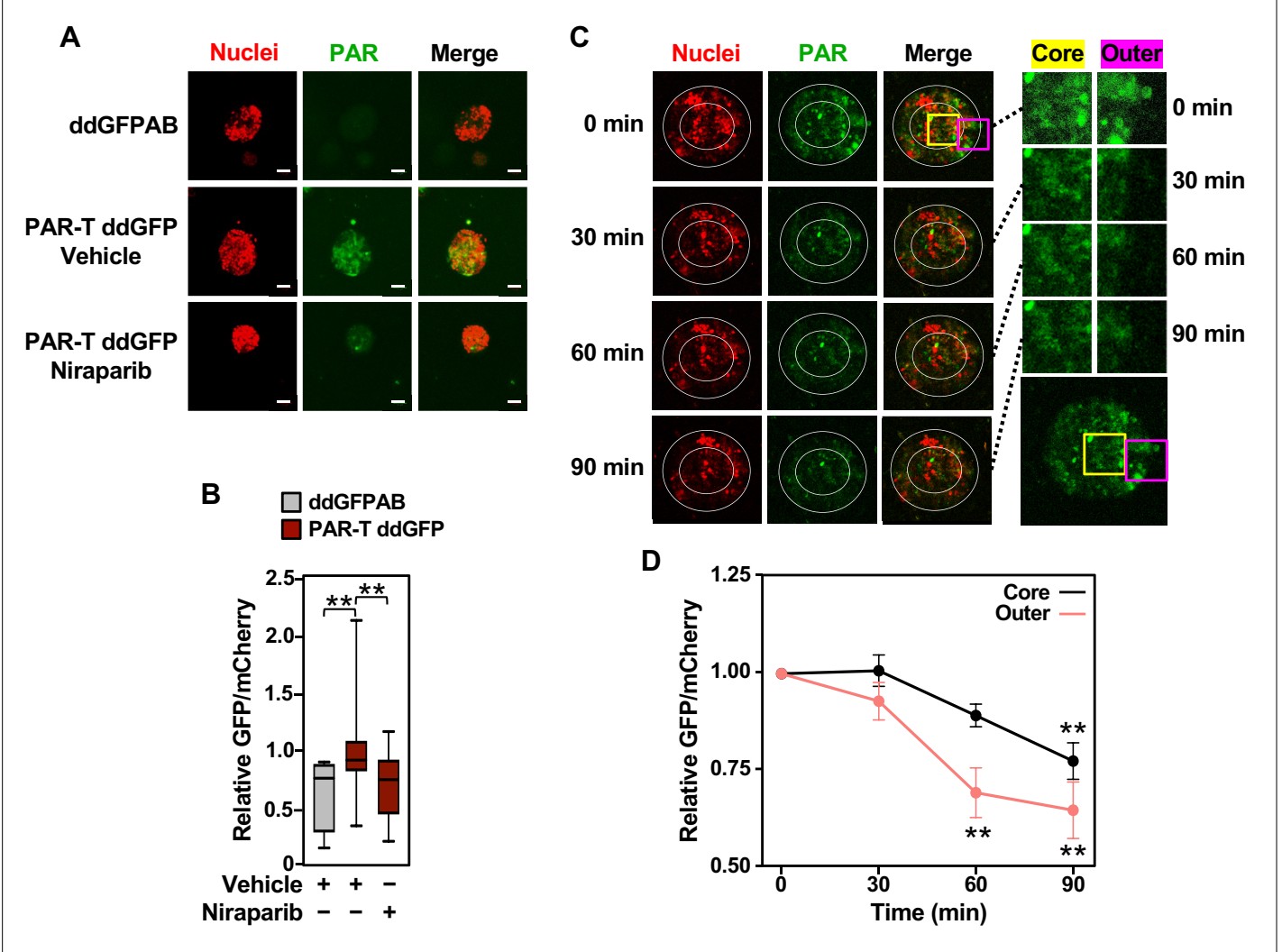

**Figure 3.** Tracking the spatial accumulation of PAR using PAR-T ddGFP. (**A, B**) Representative images and quantitative analysis of Z-projections of cancer spheroids formed using MCF-7 cells subjected to Dox-induced expression of PAR-T ddGFP. The spheroids were treated with 20 µM Niraparib for 24 hr prior to imaging. The scale bar is 50 µm. Each bar in the graph in (**B**) represents the mean ± SEM of the relative levels of the fluorescence intensity (n=3 biological replicates containing a total of 8 spheroids for the ddGFPAB control and at least 19 spheroids for PAR-T ddGFP, one-way ANOVA, *p<0.05). (**C, D**) Representative images (**C**) of Z-projections of cancer spheroids formed using MCF-7 cells subjected to Dox-induced expression of the PAR-T ddGFP. The spheroids were treated with 20 µM Niraparib and live-cell imaging was performed at the indicated times. (*Left*) The spheroids were divided into 'outer' and 'core' sections for quantification as indicated by the white circles. (*Right*) Enlargement of the indicated areas from the left panels (yellow, core; pink, outer) as indicated. Each point in the graph in (**D**) represents the mean ± SEM of the relative levels of PAR-T ddGFP fluorescence intensity normalized to mCherry (n=5, one-way ANOVA, *p<0.05 and **p<0.01).

*supplement 1B-D*). Luminescence from an unsplit Firefly luciferase remained unaltered with these treatments (*Figure 4—figure supplement 1D and E*).

Although split firefly luciferase-based approaches were capable of detecting signals from cells in vitro, the signal intensity from the split firefly luciferase-based PAR-T sensors were ~10,000-fold less than intact firefly luciferase, which makes it difficult to use this sensor in vivo. This limitation could be due to the bulkiness of firefly luciferase, which may interfere with the function of the domains fused to them in complementation assays (*Wang et al., 2020*; *Yano et al., 2018*). Therefore, we decided to employ a Nano luciferase (NanoLuc)-based split luciferase complementation system (*Dixon et al., 2016*; *Hall et al., 2012*), which is a smaller, brighter, and more stable luciferase compared to firefly luciferase (*Figure 4A*).

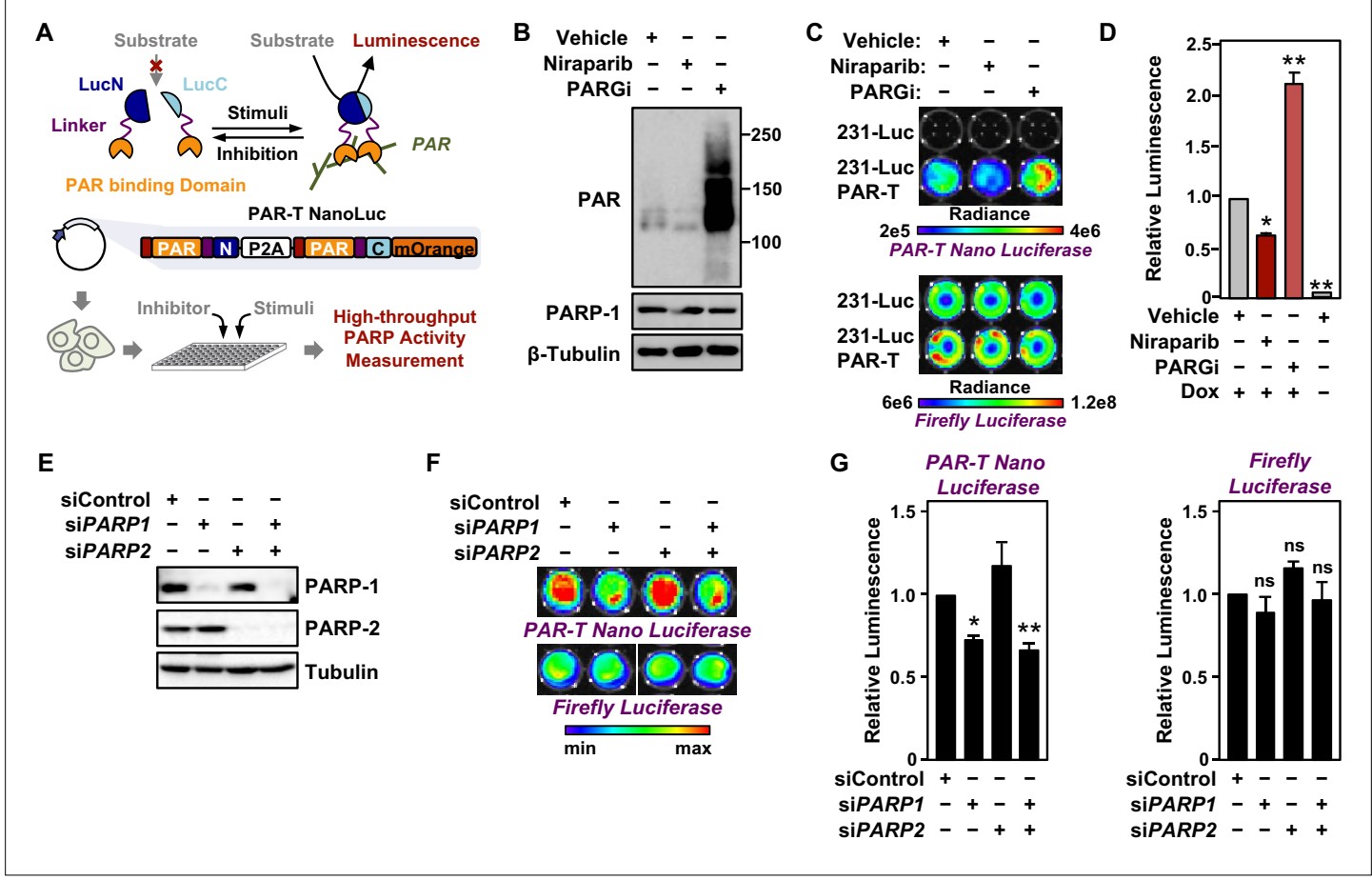

**Figure 4.** Tracking PAR accumulation in response to PARP or PARG inhibition, or *PARP1* or *PARP2* knockdown using PAR-T NanoLuc. (**A**) Schematic diagram of the plasmid constructs used to express the split Nano luciferase PAR-Tracker (PAR-T NanoLuc) in mammalian cells. The constructs contain DNA segments encoding (1) Flag tag (red), (2) ADP-ribose binding domain (yellow), (3) a flexible linker (purple), and (4) the N-terminal (dark blue) or C-terminal (light blue) fragments of NanoLuc. (**B**) Western blot analysis of 231-PAR-T NanoLuc cells treated with Niraparib or PARG inhibitor as indicated. (**C, D**) Bioluminescence imaging (**B**) of MDA-MB-231-luc cells subjected to Dox-induced expression of PAR-T NanoLuc (231-PAR-T NanoLuc). The cells were treated with 20 µM Niraparib or 20 µM PARG inhibitor (PDD00017273) for 2 hr prior to bioluminescence imaging. Each bar in the graph in (**D**) represents the mean ± SEM of the relative levels of the ratio of luminescence of NanoLuc to firefly luciferase (n=3, two-way ANOVA, *p<0.01 and **p<0.0001). (**E**) Western blot analysis of MDA-MB-231-luc cells subjected to siRNA mediated knockdown of *PARP1* or *PARP2* as indicated. (**F, G**) Bioluminescence measurement (**F**) of 231-PAR-T NanoLuc cells subjected to *PARP1* or *PARP2* knockdown. Each bar in the graph in (**G**) represents the mean ± SEM of the relative levels of luminescence of NanoLuc (*left*) or firefly luciferase (*right*) (n=3, t-test, *p<0.05 and **p<0.01; ns=not significant).

The online version of this article includes the following figure supplement(s) for figure 4:

**Figure supplement 1.** Characterization of a luminescence-based PAR-Tracker.

**Figure supplement 2.** Measuring the levels of PAR in a limited number of cells using PAR-T NanoLuc.

To quantitatively assess the activity of this luminescent PAR-Tracker (PAR-T NanoLuc), we expressed it in a Dox-dependent manner in human breast cancer cells that also stably express firefly luciferase (MDA-MB-231-Luc cells) (*Figure 4A*). In this way, we had an internal standard (i.e., the signal from the firefly luciferase), which allowed us to account for changes in cell viability or tumor size in these experiments. We first tested if there was cross-reactivity of the two luciferases (NanoLuc and firefly luciferase) to the substrates; we observed specific detection of firefly luciferase with D-Luciferin and NanoLuc with furimazine with no cross-reactivity (*Figure 4B-D*). Moreover, the luminescence of PAR-T NanoLuc is only 30-fold lower than intact firefly luciferase (*Figure 4C*). PARP-1 depletion reduced the luminescence from PAR-T NanoLuc with little effect on the luminescence of firefly luciferase (*Figure 4E-G*). Interestingly, knockdown of PARP-2 had no effect on luminescence from PAR-T NanoLuc. Nevertheless, the luminescent PAR-T NanoLuc sensor is highly sensitive and can be used to detect PAR in

1000 cells with a dynamic range of approximately threefold (minimum to maximum) (*Figure 4—figure supplement 2*).

DNA damaging agents, such as UV irradiation and gamma irradiation, activate PARP-1 and promote auto and trans PARylation of PARP-1 and other DNA damage repair proteins, respectively, that are recruited to sites of DNA damage (*Ray Chaudhuri and Nussenzweig, 2017*). Since the PAR-T sensor can detect $H_2O_2$-induced PARP-1 activation (*Figure 2*), we assessed whether it can detect radiation-induced PARP-1 activation. We subjected the MDA-MB-231-Luc cells to Dox-induced expression of PAR-T NanoLuc and then exposed the cells to UV radiation. We observed that UV radiation-induced PARP-1 activation as assessed by an accumulation of PAR on Western blots (*Figure 5A*). This was further enhanced by inhibition of PARG, whereas inhibition of PARP-1 blocked the UV-induced PARP-1 activation (*Figure 5A*). UV radiation of PARG inhibitor-treated cells enhanced PAR-T luminescence, whereas UV radiation of PARP inhibitor-treated cells reduced the PAR-T luminescence (*Figure 5B and C*). None of these treatments affected the luminescence from firefly luciferase (*Figure 5B and C*).

In a similar manner, we also performed a time course of UV-mediated PARP-1 activation using live cell luminescence assay with the PAR-T NanoLuc sensor. We subjected the MDA-MB-231 cells to Dox-induced expression of PAR-T NanoLuc or intact Nano luciferase and then exposed the cells to UV radiation. Consistent with the previous experiment, we observed a time-dependent increase in PAR-T NanoLuc signal in vehicle-treated cells, but not in Niraparib-treated cells. Interestingly, UV-mediated increases in PARP-1 activation were more spontaneous in PARG inhibitor-treated cells (*Figure 5D*). The PAR levels under basal (–UV) conditions were low, resulting in only a 50% decrease in PAR-T NanoLuc signal with Niraparib treatment (*Figure 5A-C*). The decrease in PAR-T NanoLuc signal was greater when UV-treated cells were pre-treated with Niraparib, which is consistent with the results from Western blot analysis (*Figure 5A*).

## Comparison of assay performance using the PAR-T sensor and conventional PAR detection reagents

Next, we performed a set of assays to compare the performance of the PAR-T sensors to conventional PAR detection reagents (WWE-Fc and PAR antibody) in a variety of assays. In sum, we compared: (1) Western blotting with WWE-Fc versus fluorescence assay with PAR-T ddGFP, which were performed in conjunction with ARH3-mediated degradation of PAR in vitro (*Figure 6A*); (2) Western blotting with WWE-Fc versus live-cell luciferase assay using PAR-T NanoLuc, which were performed in conjunction with UV-induced DNA damage in MDA-MB-231 cells (*Figure 6B*); (3) enzyme-linked immunosorbent assay (ELISA) with PAR antibody versus fluorescence assay with PAR-T ddGFP, which were performed using immobilized PAR (*Figure 6C*); and (4) immunofluorescence with WWE-Fc versus live-cell imaging using PAR-T ddGFP, which were performed using $H_2O_2$-mediated PARP-1 activation in 293T cells (*Figure 6D and E*).

We compared the dynamic ranges of the PAR-T sensors with the other reagents (WWE-Fc and PAR antibody) in various PAR detection assays, such as Western blotting and ELISA (*Figure 6F*). We found that Western blotting with WWE-Fc had the highest dynamic range for detection of PAR (eightfold), but the dynamic range of live-cell luciferase assay with PAR-T NanoLuc was comparable (sixfold) (*Figure 6F*). While PAR-T ddGFP in a modified fluorescence assay had a larger dynamic range than PAR antibody in an ELISA (6-fold vs. 3.5-fold) when the assays were performed using immobilized PAR, it had a lower dynamic range when used for live-cell imaging (4.4-fold). This can be explained, in part, by the higher autofluorescence of cells, which can diminish the dynamic range of the PAR-T ddGFP sensors. Thus, this sensor requires further optimization to increase the signal-noise ratio. Nevertheless, the performance of PAR-T ddGFP in live cells is comparable to that of an immunofluorescence assay with WWE-Fc (4.4-fold vs. 5-fold).

## Detection of PAR production from PARP-1 activation under physiological conditions

We previously showed that PARP-1 catalytic activity decreases during the initial differentiation of preadipocytes (*Huang et al., 2020*; *Luo et al., 2017*; *Ryu et al., 2018*). Thus, adipogenesis is a unique biological process to study the dynamics of PAR accumulation from changes in PARP-1 activity under physiological conditions. We used the PAR-T NanoLuc sensor to investigate changes in PARP-1 activity during early adipogenesis of murine preadipocytes (i.e., 3T3-L1 cells). We observed a decrease

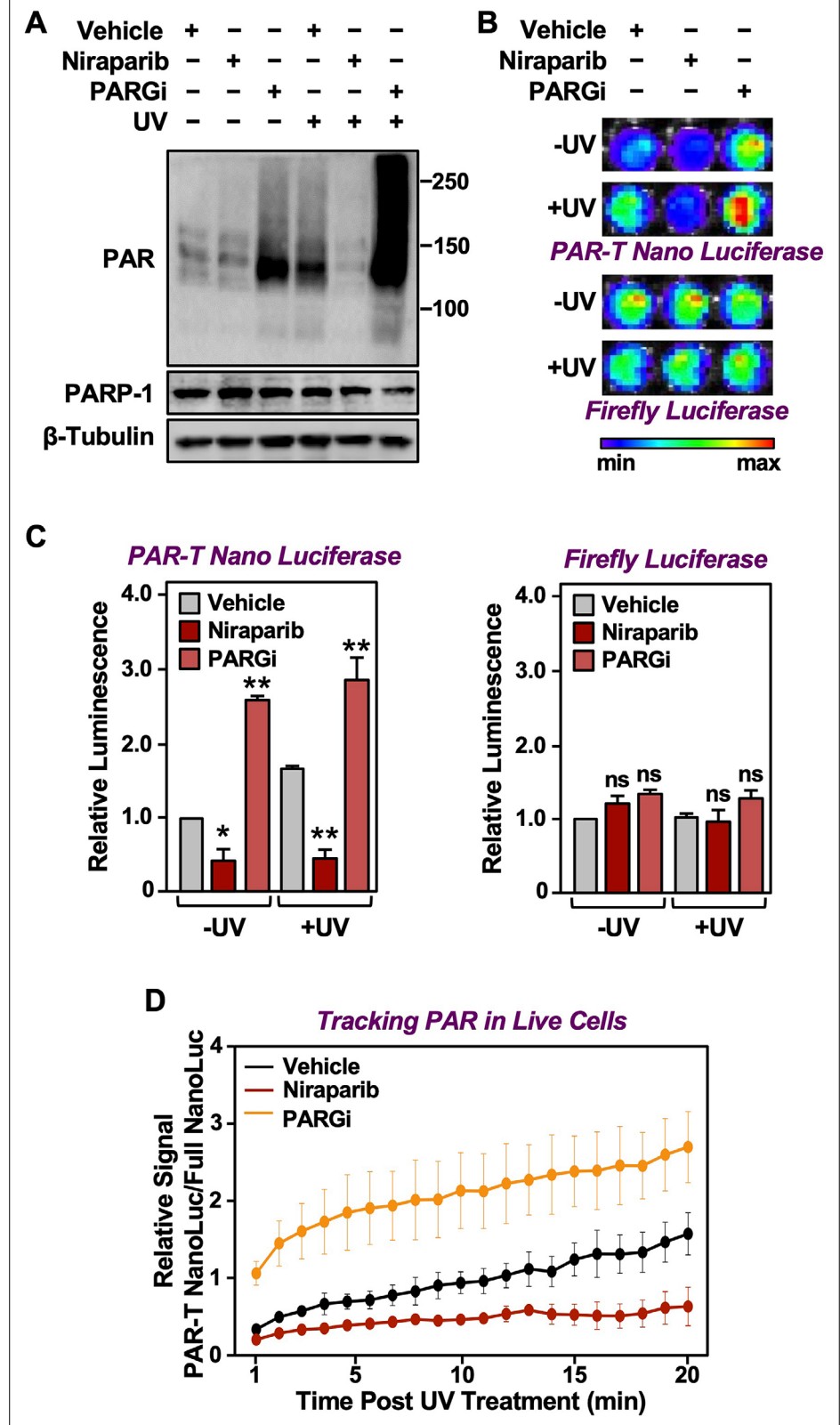

**Figure 5.** Tracking PAR accumulation in response to UV-induced DNA damage using PAR-T NanoLuc. (**A**) Western blot analysis of 231-PAR-T NanoLuc cells treated with Niraparib or PARG inhibitor prior to UV radiation. (**B, C**) Bioluminescence imaging (**B**) of 231-PAR-T NanoLuc cells treated with 20 μM Niraparib or 20 μM PARG inhibitor for 2 hr prior to UV radiation. Each bar in the graph in (**C**) represents the mean ± SEM of the relative levels of

*Figure 5 continued on next page*

Figure 5 continued

luminescence with NanoLuc (*left*) or firefly luciferase (*right*) (n=3, one-way ANOVA, *p<0.05, **p<0.001; ns=not significant). (**D**) Time course of Bioluminescence imaging of 231-PAR-T NanoLuc and 231-Full Nano luciferase cells treated with 20 µM Niraparib or 20 µM PARG inhibitor for 2 hr prior to UV radiation. Each point in the graph represents the mean ± SEM of the relative levels of luminescence from PAR-T NanoLuc normalized to full Nano luciferase (n=3).

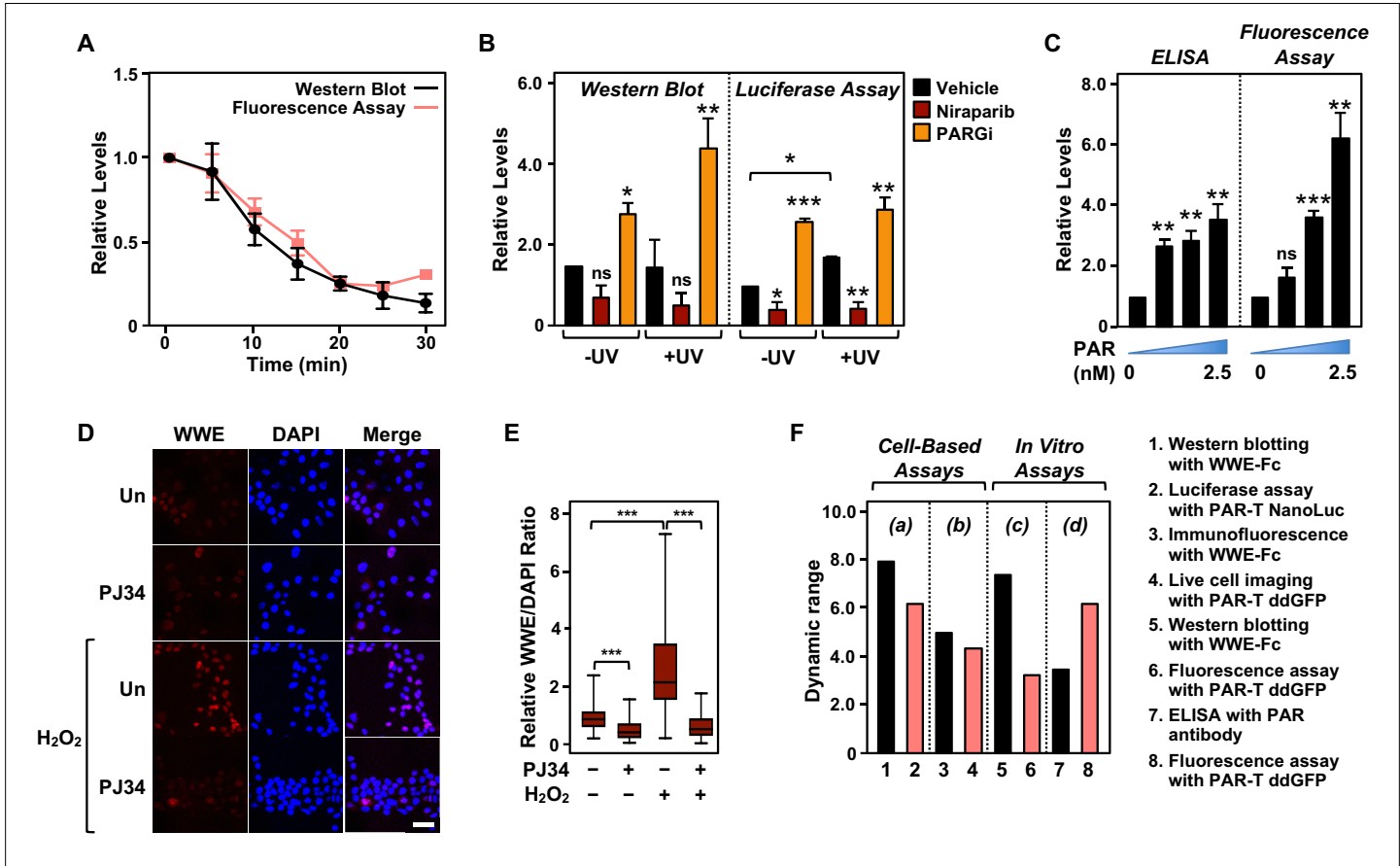

**Figure 6.** Comparison of assay performance using the PAR-T sensor and conventional PAR detection reagents. (**A**) Quantitative analysis of Western blot analysis and fluorescence measurements (shown in *Figure 1G*) of the time course of in vitro PAR degradation using recombinant ARH3. Each line plot in the graph represents mean ± SEM of relative intensities (n=3). (**B**) Quantitative analysis of Western blot analysis and bioluminescence imaging (shown in *Figure 4B*) of 231-PAR-T NanoLuc cells treated with 20 µM Niraparib or 20 µM PARG inhibitor for 2 hr prior to UV radiation. Each bar in the graph represents the mean ± SEM of the relative intensities (n=3, one-way ANOVA, *p<0.05, **p<0.001, and ***p<0.0001; ns=not significant). (**C**) Measurements of ELISA and fluorescence intensities using 0, 0.625, 1.25, and 2.5 nM concentrations of purified PAR. Each bar in the graph in represents the mean ± SEM of the relative intensities (n=3, paired t-test, *p<0.05, **p<0.01, and ***p<0.001; ns=not significant). (**D**) Immunofluorescence assay using WWE-Fc to measure PAR formation in response to $H_2O_2$ using 293T cells. The cells were treated with 20 µM PJ34 (vs. untreated control, 'Un') for 2 hr prior to 15 min of treatment with 1 mM $H_2O_2$. The images were collected using a confocal microscope. (**E**) Quantification of the results in (**D**). Each bar in the graph represents the mean ± SEM of the relative levels of the fluorescence intensity of PAR normalized to DAPI (n=3 biological replicates with at least 150 cells in total, one-way ANOVA, ***p<0.0001). (**F**) Representation of the dynamic ranges of PAR-T sensors in comparison to other available PAR detection tools as indicated: (a) Western blotting with WWE-Fc versus live-cell luciferase assay using PAR-T NanoLuc was performed using UV-induced DNA damage in MDA-MB-231 Luc cells (from (**B**)); (b) Immunofluorescence with WWE-Fc versus live-cell imaging using PAR-T ddGFP was performed using $H_2O_2$-mediated PARP-1 activation in 293T cells (from (**D**)); (c) Western blotting with WWE-Fc versus fluorescence assay with PAR-T ddGFP was performed using ARH3 mediated degradation of PAR in vitro (from (**A**)); (d) ELISA versus fluorescence assay with PAR-T ddGFP was performed using immobilized PAR (from (**C**)).

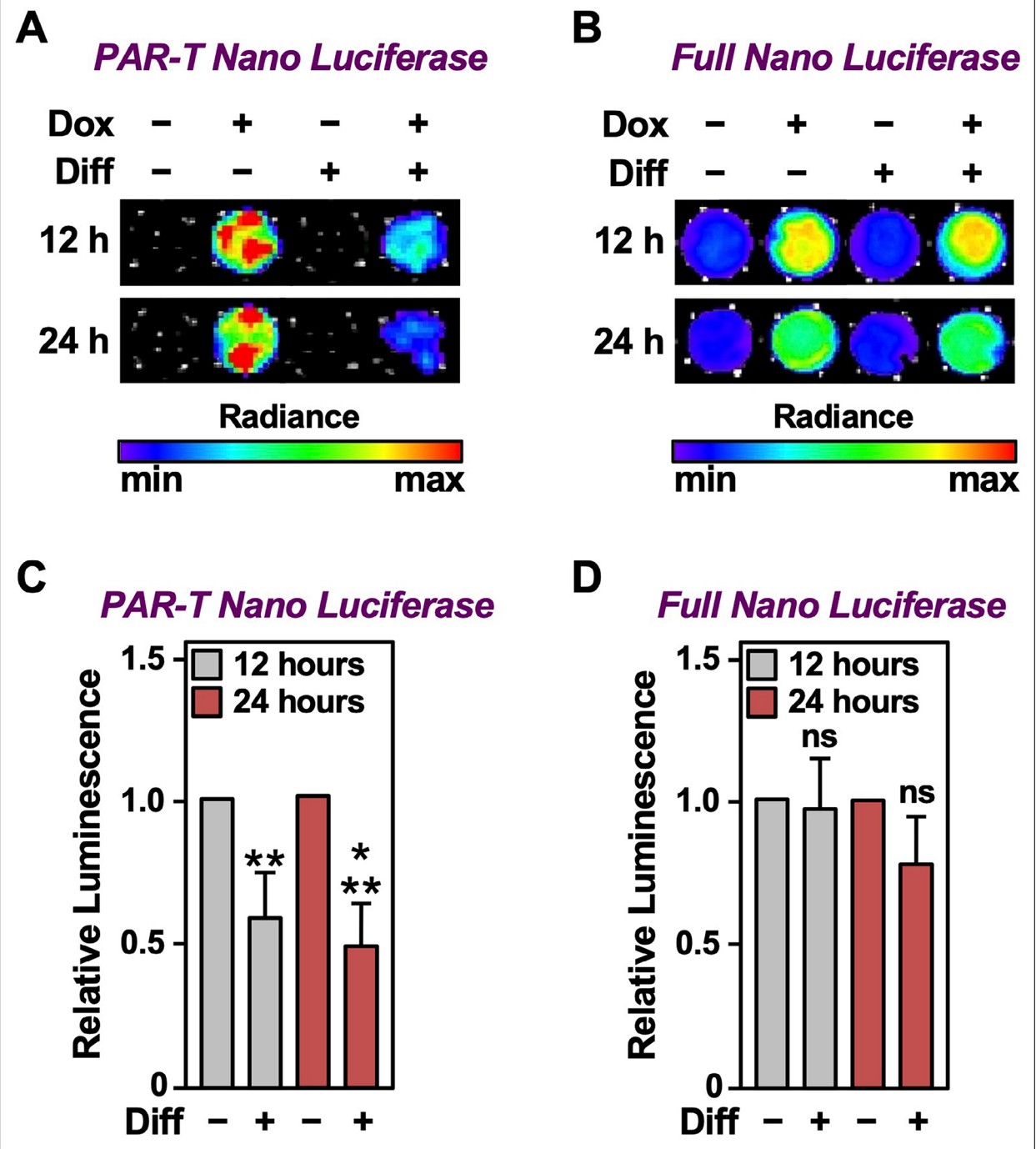

**Figure 7.** Tracking the levels of PAR during adipogenesis. (**A, B**) Bioluminescence imaging of PAR-T NanoLuc (**A**) and unsplit NanoLuc (**B**) in 3T3-L1 cells subjected to adipogenic differentiation for 12 or 24 hr. (**C, D**) Quantification of signals from PAR-T NanoLuc (**C**) and unsplit NanoLuc (**D**) during adipogenesis. Each bar in the graph represents the mean ± SEM of the relative levels of the luminescence of NanoLuc (n=4; t-test, **p<0.01 and ***p<0.001). Comparisons between experimental conditions with the intact NanoLuc are not statistically significant (ns).

in the signal from PAR-T NanoLuc by 12 hr of differentiation and a greater reduction in PAR-T NanoLuc signal by 24 hr of differentiation (*Figure 7*), consistent with our previous observation that PARP-1 activation decreases precipitously during adipogenesis (*Huang et al., 2020*; *Luo et al., 2017*; *Ryu et al., 2018*). These results further highlight the high sensitivity of PAR-T NanoLuc sensor, which can be used to study physiological changes in PAR levels during biological processes, such as adipogenesis.

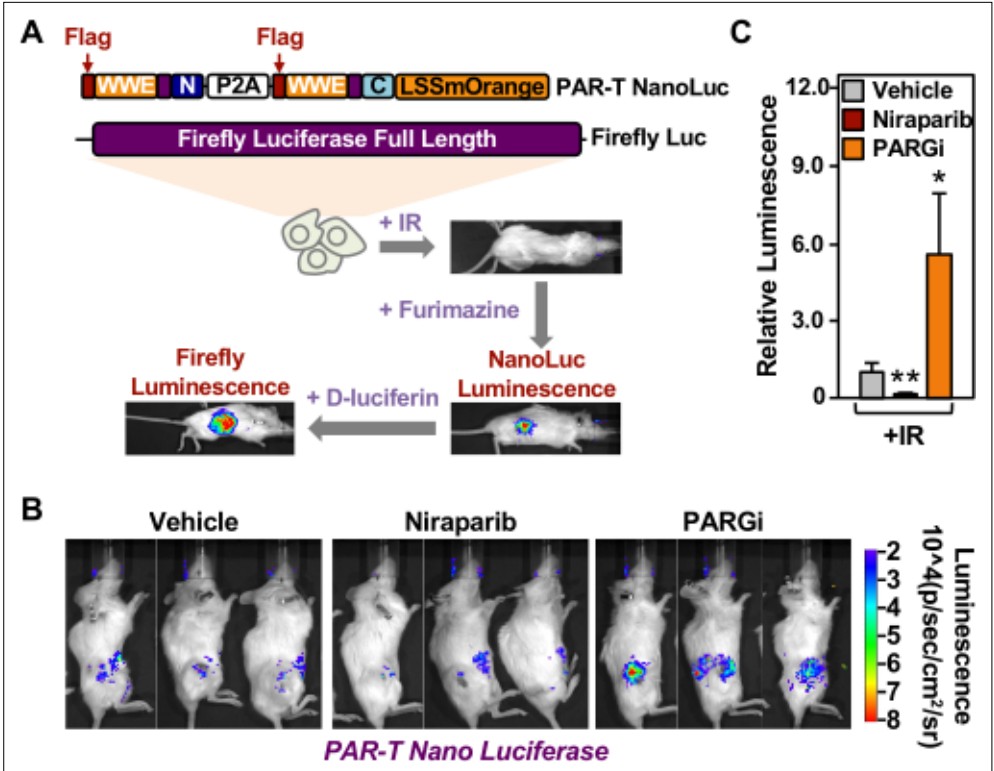

**Figure 8.** Measuring the levels of PAR in tissues in living animals using PAR-T NanoLuc. (**A**) Schematic diagram of the in vivo studies performed using 231-PAR-T NanoLuc cells. (**B, C**) Bioluminescence imaging (**B**) of xenograft tumors formed using 231-PAR-T NanoLuc cells. The tumors were subjected to 5 Gy IR radiation and treatment with PARP inhibitor (Niraparib) or PARG inhibitor as indicated prior to BLI imaging. Each bar in the graph in (**C**) represents the mean ± SEM of the relative levels of the ratio of luminescence of NanoLuc to firefly luciferase (n=6 for the vehicle and PARG inhibitor treatment cohorts and n=7 for the Niraparib treatment cohort; t-test, *p<0.05 and **p<0.01).

The online version of this article includes the following figure supplement(s) for figure 8:

**Figure supplement 1.** Tracking the levels of PAR in tissues in living animals using PAR-T NanoLuc.

## Detection of PAR production from PARP-1 activation in vivo

To assess the in vivo utility of PAR-T NanoLuc, we established xenograft tumors in NOD/SCID/gamma (NSG) mice using MDA-MB-231-Luc cells and induced the expression of PAR-T NanoLuc with Dox (*Figure 8A*). Similar to the in vitro experiments, we detected an increase in PAR levels with the PAR-T NanoLuc sensor when the gamma-irradiated mice were treated with PARG inhibitor, but the signal was decreased when the gamma-irradiated mice were treated with PARP inhibitor (*Figure 8B* and *Figure 8—figure supplement 1A*). We normalized the signal from PAR-T to luminescence from firefly luciferase to confidently measure the differences in PAR levels, while accounting for the variability in tumor sizes (*Figure 8C*).

We also measured PAR accumulation in breast cancer cells injected into C57/BL6 mice without establishing xenograft tumors over a 24-hr time course post injection, with or without PARG inhibitor treatment (*Figure 8—figure supplement 1B-D*). PAR accumulation was readily detected in the breast cancer cells injected into the mice in the absence of treatment. Upon treatment with PARG inhibitor, the luminescence from PAR-T NanoLuc increased significantly by 6 hr and then diminished by 24 hr. These results demonstrate that the PAR-T NanoLuc sensor has sufficient sensitivity to detect dynamic changes in PAR production in tissues of living animals in vivo.

## Discussion

Naturally occurring ARBDs have been invaluable tools for developing novel ADPR detection reagents and sensors (*Forst et al., 2013*; *Gibson et al., 2017*; *Timinszky et al., 2009*). In this study, we developed a set of PAR sensors that are useful tools for in vitro assays, live cells, and tissues in living animals (*Table 1*). To this end, we constructed a ddGFP-based fluorescent PAR-Tracker (PAR-T) that can be used for in vitro assays and live-cell imaging, to track dynamic changes in PAR levels in a single cell (*Figure 2*) and spatially resolve PAR accumulation in tissue-like spheroids (*Figure 3*). In addition, we made a split NanoLuc-based luminescent PAR-T containing LSSmOrange that can be used to track physiological changes in PAR levels during differentiation (*Figure 7*). It is extremely sensitive and can be used to detect PAR in xenograft tumors in living mice (*Figure 8*).

### Previously developed PAR sensors

In previous work, *Furman et al., 2011* made a bivalent split-protein PAR-specific sensor encompassing the PBZ modules of APLF (residues 376–441) (*Ahel et al., 2008*) attached to each half of split firefly luciferase (split-Fluc) (*Furman et al., 2011*). This tool allows the detection of PAR from biochemical reactions and cell lysates (*Furman et al., 2011*). *Krastev et al., 2018* also made an APLF PBZ domain-fused split fluorescence sensor based on Venus fluorescent protein to detect PAR (*Krastev et al., 2018*). In a recent study, *Serebrovskaya et al., 2020* developed FRET-based PAR sensors by fusing the WWE domain from RNF146 to the Torquoise2 and Venus fluorescent proteins (*Serebrovskaya et al., 2020*). These sensors detect PAR formation in live cells (*Krastev et al., 2018*; *Serebrovskaya et al., 2020*).

However, the aforementioned approaches for PAR sensors have the following limitations. (1) they are irreversible, thus limiting their use for measuring dynamic changes in PAR levels, (2) the requirement for measuring filtered light emissions in FRET-based sensors results in low signal intensities and narrow dynamic ranges that limit their utility in a variety of applications, (3) PBZ domains detect branched PAR chains synthesized predominantly by PARP-2 (*Chen et al., 2018*), and (4) they are unable to measure PAR levels in vivo. To overcome these limitations, we developed a set of PAR sensors in which ARBDs are fused to ddGFP or split luciferase. Assembly of dimerization-dependent fluorophores and split luciferase proteins are reversible and have a higher signal intensity (*Alford et al., 2012*; *Luker et al., 2004*). In our screen for the best ARBDs for PAR detection, the WWE domain from RNF146 and the macrodomain from AF1521 exhibited superior detection of PAR production by activated PARP-1, compared to the PBZ domains from APLF or the macro H2A.1 macrodomain (*Figure 1—figure supplement 1E* and *Table 1*). Thus, although useful, the currently available PAR sensors have multiple opportunities for improvement.

### Detection of PAR in tissues in living animals

A major goal of this work was to generate a sensor with sufficient stability and sensitivity to allow detection of PAR in tissues in living animals. We faced several challenges in doing so. For example, fluorescent sensors are not optimal for use in vivo due to high auto-fluorescence of tissue; thus, we had to use a luminescent sensor to increase the sensitivity of PAR detection in vivo. Although intact luciferase can be used in cells as reporter, achieving usable signals from split luciferase is technically challenging because (1) it is difficult to express, (2) the luminescence of split luciferase is typically 100- to 1000-fold less than intact luciferase, and (3) the wavelength emitted by luciferase exhibits poor penetration in tissues. A broadly useful PAR detection tool would need to overcome these limitations and allow real-time dynamic observations in cells and detection in tissues.

To this end, we optimized several aspects of the sensor to achieve the highest sensitivity: (1) we used NanoLuc, the smallest and brightest luciferase available (*Wang et al., 2020*), (2) we added LSSmOrange to stabilize the C-terminal fragment of NanoLuc (*Schaub et al., 2015*), (3) we used the Nano-Glo live cell substrate to be able to perform these assays in live cells, (4) we used Dox-inducible constructs to avoid any effects of expression of these constructs on cell viability, and (5) we developed a dual luciferase assay to quantify PAR levels more accurately. The blue-shifted emission of NanoLuc (at 460 nm) diffuses rapidly in tissues, and hence, it is not optimal for imaging deep tissues (*Schaub et al., 2015*). Therefore, we fused the C-terminal domain of NanoLuc with LSSmOrange fluorescent protein (red-shifted GFP variant), which has a higher fluorescence quantum yield that is compatible for excitation by the light emitted by NanoLuc. Upon addition of the luciferase substrate,

the bioluminescence energy from NanoLuc excites the LSSmOrange fluorophore, shifting the emission energy to 570 nm via BRET reaction (*Schaub et al., 2015*).

In summary, we generated a set of PAR-Trackers with significant improvements in functionality over our previous detection reagents. The PAR-T NanoLuc sensor can detect PAR levels in as few as 1000 cells with a good dynamic range of detection (*Figure 4—figure supplement 2*), and it can also detect PAR in tissues in living animals (*Figure 7*).

## Perspective and future advances for PAR sensors

Current studies on PARylation are limited to in vitro biochemical assays and end-point cellular assays. Moreover, the techniques routinely used to measure PAR levels require laborious and time-consuming assays, such as Western blotting, ELISA, immunofluorescence, or immunohistochemistry. The high sensitivity and low signal-to-noise ratios of the PAR-Trackers described here enable spatial and temporal monitoring of PAR levels in cells and in animals. Moreover, these techniques do not require exogenous ligands and involve limited manipulation to cells that can limit artifacts caused by sample handling, such as lysing or fixing of cells. Generating animal models with tissue-specific expression of the PAR-T NanoLuc sensor will enable monitoring PAR levels in specific cell types in vivo.

Although the dynamic range of the PAR-T sensors is comparable to other available PAR detection tools, the sensors can be improved further by optimizing the assay conditions and the design of the PAR-T sensors. For example, our data suggest that in plate-based fluorescence assays, immobilizing PAR on the well increases the sensitivity of detection by PAR-T ddGFP. The reduced dynamic range of PAR-T ddGFP in live-cell imaging assays can be attributed to high background autofluorescence from the cells and culture medium. To minimize this background signal, future PAR-T fluorescent sensors should be developed using fluorophores that have low background signal, such as those that emit fluorescence at a far-red wavelength. In addition, the ddGFP-A fragment has low fluorescence intensity, possibly contributing to reduced dynamic range. To avoid this, an optimized split-fluorescence sensor should be used. Similarly, the background signals contributing to reduced dynamic range in the assays using the PAR-T NanoLuc sensor can be reduced by optimizing the luciferase-fluorophore reporter pair used in the sensor.

# Materials and methods

**Key resources table**

| Reagent type (species) or resource | Designation | Source or reference | Identifiers | Additional information |
|---|---|---|---|---|
| Strain, strain background (*Escherichia coli*) | BL21(DE3-pLysis) | Thermo Fisher Scientific | Cat. no. C606010 | |
| Cell line (*Homo sapiens*) | MCF-7 (female adult breast cancer) | ATCC | RRID:CVCL_0031 | |
| Cell line (*H. sapiens*) | 3T3-L1 (male adult preadipocyte) | ATCC | RRID:CVCL_0123 | |
| Cell line (*H. sapiens*) | HeLa (female adult cervical cancer) | ATCC | RRID:CVCL_0030 | |
| Cell line (*H. sapiens*) | MDA-MB-231 Luc (female adult breast cancer) | Obtained from Dr. Srinivas Malladi, UT Southwestern | | |
| Cell line (*H. sapiens*) | HEK 293T (normal embryonic kidney) | ATCC | RRID:CVCL_0063 | |
| Transfected construct (*H. sapiens*) | *PARP1* siRNA | Sigma-Aldrich | Cat. no. SASI_Hs01_0033277 | |
| Transfected construct (*H. sapiens*) | *PARP2* siRNA | Sigma-Aldrich | Cat. no. SASI_Hs01_0013-1488 | |
| Antibody | Anti-poly-ADP-ribose binding reagent (rabbit monoclonal; IgG Fc) | Millipore *Gibson et al., 2017* | Cat. no. MABE1013 | WB (5 µg/ml) IF (1:500) |
| Antibody | PARP-1 (rabbit polyclonal) | Active Motif | Cat. no. 39559 | WB (1:1000) |
| Antibody | β-tubulin (rabbit polyclonal) | Abcam | Cat. no. ab6046 RRID: AB_2210370 | WB (1:1000) |
| Antibody | HRP-conjugated anti-rabbit IgG (goat polyclonal) | Pierce | Cat. no. 31460 RRID: AB_228341 | WB (1:5000) |

*Continued on next page*

| Reagent type (species) or resource | Designation | Source or reference | Identifiers | Additional information |
|---|---|---|---|---|
| Antibody | Alexa Fluor 594-conjugated anti-rabbit IgG (donkey polyclonal) | Thermo Fisher Scientific | Cat. no. A-21207 RRID: AB_141637 | IF (1:500) |
| Commercial assay or kit | Nano-Glo Live Cell Assay System | Promega | Cat. no. N2011 | |
| Commercial assay or kit | Nano-Glo Luciferase Assay System | Promega | Cat. no. N1110 | |
| Commercial assay or kit | Luciferase Assay System | Promega | Cat. no. E1500 | |
| Commercial assay or kit | PAR ELISA Kit | Cell Biolabs | Cat. no. XDN-5114 | |
| Chemical compound, drug | PJ34 | Enzo | Cat. no. ALX-270 | In cells (20 µM) |
| Chemical compound, drug | Niraparib | MedChem Express | Cat. no. HY-10619 | In cells (20 µM) In vivo (25 mg/kg body weight) |
| Chemical compound, drug | PDD00017273 | MedChem Express | Cat. no. HY-108360 | In cells (20 µM) In vivo (25 mg/kg body weight) |

## Cell culture and treatments

HeLa, 293T, 3T3-L1, and MCF-7 cells were obtained from the American Type Cell Culture, and MDA-MB-231-Luc cells were obtained from Dr. Srinivas Malladi, UT Southwestern Medical Center. Fresh cell stocks were regularly replenished from the original stocks, verified for cell-type identity using the GenePrint 24 system (Promega, B1870), and confirmed as mycoplasma-free using a commercial testing kit every 3 months.

HeLa, 293T, and MCF-7 cells were cultured in DMEM (Sigma-Aldrich, D5796) supplemented with 10% fetal bovine serum (FBS; Sigma-Aldrich, F8067) and 1% penicillin/streptomycin. 3T3-L1 cells were cultured in DMEM (Cellgro, 10-017 CM) supplemented with 10% FBS (Atlanta Biologicals, S11550) and 1% penicillin/streptomycin. For the luciferase assays, 5000 3T3-L1 cells were plated in each well of a 96-well format plate. For the induction of adipogenesis, the 3T3-L1 cells were grown to confluence and then cultured for 2 more days until contact inhibited. The cells were then treated for 2 days with MDI adipogenic cocktail containing 0.25 mM IBMX, 1 µM dexamethasone, and 10 µg/ml insulin for 12 or 24 hr as indicated. Expression of PAR-T NanoLuc was induced by treating the 3T3-L1 cells with doxycycline (Dox) for 24 hr before the luciferase assay was performed.

In some cases, the cells were treated with various inhibitors as described herein. For inhibition of nuclear PARPs, the cells were treated with PJ-34 (20 µM; Enzo, ALX-270) or Niraparib (20 µM; MedChem Express, HY-10619) for 2 hr. For inhibition of PARG, the cells were treated with PDD00017273 (20 µM; MedChem Express, HY-108360) for 2 hr. For UV-induced DNA damage, the cells were treated with 50 mJ/cm$^2$ UV irradiation for 15 min.

## Vectors for ectopic expression and knockdown

The vectors described below were generated using the oligonucleotide primers described in the next section. All constructs were verified by sequencing.

### Mammalian expression vectors

The plasmids for Dox-inducible expression of the ddGFP PAR-T constructs were generated using a cDNA for ddGFP-A (Addgene, 40286) or ddGFP-B (Addgene, 40287). cDNAs for the PAR binding domains were amplified from previously published pET19b constructs (*Gibson et al., 2017*). The cDNAs were assembled and cloned first into pCDNA3 and then into pInducer20 or pET19b using Gibson assembly (NEB, E2621). The split luciferase constructs were synthesized as gene blocks (Integrated DNA Technologies), and then cloned into the pInducer20 vectors using Gibson assembly.

## List of oligonucleotide primers used for cloning

DNA sequence encoding the WWE domain used in this manuscript

GGAAATGGTGAATATGCATGGTATTATGAAGGAAGAAATGGGTGGTGGCAGTACGATGAG
CGCACTAGTAGAGAGCTGGAAGATGCTTTTTCCAAAGGTAAAAAGAACACTGAAATGTTA
ATTGCTGGCTTTCTGTATGTCGCTGATCTTGAAAACATGGTTCAATATAGGAGAAATGAA
CATGGACGTCGCAGGAAGATTAAGCGAGATATAATAGATATACCAAAGAAGGGAGTAGCT
GGACTTAGG

Corresponding amino acid sequence

GNGEYAWYYEGRNGWWQYDERTSRELEDAFSKGKKNTEMLIAGFLYVADLENMVQYRRNE
HGRRRKIKRDIIDIPKKGVAGLR

Primers for cloning ddGFPA-ddGFPB into pCDNA3

Forward 1: 5'-AGGGGCGGAATTCCTCTAGTTCAATGCCCCAGGTGGTG -3'
Reverse 1: 5'-AGGGGCGGAATTCCTCTAGTTCAATGCCCCAGGTGGTG -3'
Forward 2: 5'-ATTACGCTCTTGAAGCAACCATGGCCACCATCAAAGAGTTCATGC -3'
Reverse 2: 5'-TAGGGCCCTCTAGATGCATGTTACTTGTACCGCTCGTC -3'

Primers for cloning WWE-ddGFPA and WWE-ddGFPB into pCDNA3

Forward 1: 5'-ATGACAAGCTTGAAGCAACCGGAAATGGTGAATATGCATGGTATTATG -3'
Reverse 1: 5'-AGGGGCGGAATTCCTCTAGTTCAATGCCCCAGGTGGTG -3'
Forward 2: 5'-ATTACGCTCTTGAAGCAACCGGAAATGGTGAATATGCATG-3'
Reverse 2: 5'-TAGGGCCCTCTAGATGCATGTTACTTGTACCGCTCGTC -3'

Primers for cloning ddGFPA or ddGFPB into pET19b

pET19b-ddGFPA Forward:
5'-TATCGACGACGACGACAAGCATATGCTCGAGATGGCGAGCAAGAGCGAG -3'
pET19b-ddGFPA Reverse:
5'-TCGGGCTTTGTTAGCAGCCGGATCCTCAATGCCCCAGGTGGTG-3'
pET19b-ddGFPB Forward:
5'-TATCGACGACGACGACAAGCATATGCTCGAGACCATCAAAGAGTTCATGC -3'
pET19b-ddGFPB Reverse:
5'-TCGGGCTTTGTTAGCAGCCGGATCCTTACTTGTACCGCTCGTC -3'

Primers for cloning WWE-ddGFPA or WWE-ddGFPB into pET19b

pET19b-WWE-ddGFPA Forward:
5'-TATCGACGACGACGACAAGCATATGCTCGAGGGAAATGGTGAATATGCATG -3'
pET19b-WWE- ddGFPA Reverse:
5'-TCGGGCTTTGTTAGCAGCCGGATCCTCAATGCCCCAGGTGGTG -3'
pET19b-WWE-ddGFPB Forward:
5'-TATCGACGACGACGACAAGCATATGCTCGAGGGAAATGGTGAATATGCATG -3'
pET19b-WWE- ddGFPB Reverse:
5'-TCGGGCTTTGTTAGCAGCCGGATCCTTACTTGTACCGCTCGTC -3'

Primers for cloning MacroH2A.1-ddGFPA or MacroH2A.1-ddGFPB into pET19b

Forward 1:
5'-TATCGACGACGACGACAAGCATATGCTCGAGGGTGAAGTCAGTAAGGCAGC -3'
Reverse 1:
5'-AGAATTCTAGGTTGGCGTCCAGCTTGGC-3'
pET19b-MacroH2A.1-ddGFPA Forward:

5′-GGACGCCAACCTAGAATTCTCGACAGGGCATG-3′
pET19b-MacroH2A.1-ddGFPA Reverse:
5′-TCGGGCTTTGTTAGCAGCCGGATCCTCAATGCCCCAGGTGGTG -3′
pET19b-MacroH2A.1-ddGFPB Forward:
5′-GGACGCCAACCTAGAATTCTCGACAGGG -3′
pET19b-MacroH2A.1-ddGFPB Reverse:
5′-TCGGGCTTTGTTAGCAGCCGGATCCTTACTTGTACCGCTCGTC -3′

## Primers for cloning PBZ-ddGFPA or PBZ-ddGFPB into pET19b

Forward 1:
5′-TATCGACGACGACGACAAGCATATGCTCGAGGATTCAGTTCTACAAGGTTC -3′
Reverse 1: 5′-AGAATTCTAGTGGAAGCGTATTATGTCTATATTC -3′
pET19b-PBZ-ddGFPA Forward:
5′-TACGCTTCCACTAGAATTCTCGACAGGGCATG -3′
pET19b-PBZ-ddGFPA Reverse:
5′-TCGGGCTTTGTTAGCAGCCGGATCCTCAATGCCCCAGGTGGTG-3′
pET19b-PBZ-ddGFPB Forward:
5′-TACGCTTCCACTAGAATTCTCGACAGGG-3′
pET19b-PBZ-ddGFPB Reverse:
5′-TCGGGCTTTGTTAGCAGCCGGATCCTTACTTGTACCGCTCGTC -3′

## Primers for cloning MacroAF-ddGFPA or PBZ-ddGFPB into pET19b

Forward 1:
5′-TATCGACGACGACGACAAGCATATGCTCGAGATGGAACGGCGTACTTTAATC -3′
Reverse 1: 5′-AGAATTCTAGAAGACTCCTCTCAAAGAC -3′
pET19b-*MacroAF*-ddGFPA Forward:
5′- GAGGAGTCTTCTAGAATTCTCGACAGGGCATG -3′
pET19b-PBZ-ddGFPA Reverse:
5′-TCGGGCTTTGTTAGCAGCCGGATCCTCAATGCCCCAGGTGGTG -3′
pET19b-*MacroAF*-ddGFPB Forward:
5′-GAGGAGTCTTCTAGAATTCTCGACAGGG -3′
pET19b-PBZ-ddGFPB Reverse:
5′-TCGGGCTTTGTTAGCAGCCGGATCCTTACTTGTACCGCTCGTC -3′

## Primers for cloning WWE-ddGFP (PAR-T ddGFP) sensors and control ddGFP into pInducer20

Forward: 5′-TCCGCGGCCCCGAACTAGTGGCCACCATGGACTACAAG -3′
Reverse: 5′-AGAGGGGCGGAATTCCTCTAGTCTTACTTGTACCGCTCGTC -3′

## Primers for cloning AF-ddGFP sensors into pInducer20

Forward 1:
5′-TCCGCGGCCCCGAACTAGTGGCCACCATGGACTACAAGGATGACGATGACAAGC
TTGAAGCAACCATGGAACGGCGTACTTTAATCATG -3′
Reverse 1: 5′-TTCCTCTAGTTCAATGCCCCAGGTGGTG -3′
Forward 2: 5′-GGGGCATTGAACTAGAGGAATTCCGCCC -3′
Reverse 2: 5′-AGAGGGGCGGAATTCCTCTAGTCTTACTTGTACCGCTCGTC -3′

## Primers for cloning split firefly luciferase (PAR-T fLuc) sensors into pCDNA3

pCDNA3-WWE/MacroAF-LucN:
Forward 1: 5′-CAAGCTTGGTACCGAGCTCGGCCACCATGGACTACAAG-3′
Reverse 1: 5′-CCATGGATCCTGAACTACCGGTCGATTC-3′
Forward 2: 5′-CGGTAGTTCAGGATCCATGGAAGACGCC -3′
Reverse-2: 5′-AGGGCCCTCTAGATGCATGCTCACATAATCATAGGTCCTCTGAC -3′
pCDNA3-WWE/MacroAF-LucC

Forward 1: 5'-CAAGCTTGGTACCGAGCTCGGCCACCATGGACTACAAG-3'
Reverse 1: 5'-GTCCGGATCCTGAACTACCGGTCGATTC -3'
Forward 2: 5'-CGGTAGTTCAGGATCCGGACCTATGATTATG -3'
Reverse-2: 5'-AGGGCCCTCTAGATGCATGCTTACAATTTGGACTTTCCG -3'

## Primers for cloning split nano luciferase sensors (PAR-T NanoLuc) into pInducer20

Forward 1: 5'-TCCGCGGCCCCGAACTAGTGATGGACTACAAGGATGAC -3'
Reverse 1: 5'-CTCCGCTTCCACTGTTGATGGTTACTCG -3'
Forward 2: 5'-CATCAACAGTGGAAGCGGAGCCACGAAC -3'
Reverse-2: 5'-GTTTAATTAATCATTACTACTTACTTGTACAGCTCGTCCATGC -3'

## Knockdown of *PARP1 and PARP2* using siRNAs

Commercially available siRNA oligos targeting *PARP1* (Sigma-Aldrich, SASI_Hs01_0033277), *PARP2* (Sigma-Aldrich, SASI_Hs01_0013-1488), and control siRNA (Sigma-Aldrich, SIC001) were transfected at a final concentration of 30 nM using Lipofectamine RNAiMAX reagent (Invitrogen, 13778150) according to the manufacturer's instructions. All experiments were performed 48 hr after siRNA transfection.

## Generation of stable cell lines

Cells were transfected with lentiviruses for stable ectopic expression. We generated lentiviruses by transfection of the pInducer20 constructs described above, together with an expression vector for the VSV-G envelope protein (pCMV-VSV-G, Addgene plasmid no. 8454), an expression vector for GAG-Pol-Rev (psPAX2, Addgene plasmid no. 12260), and a vector to aid with translation initiation (pAdVAntage, Promega) into 293T cells using GeneJuice transfection reagent (Novagen, 70967) according to the manufacturer's protocol. The resulting viruses were used to infect HeLa, MCF-7, 3T3-L1, or MDA-MB-231 cells in the presence of 7.5 µg/ml polybrene 24 and 48 hr, respectively, after initial 293T transfection. Stably transduced cells were selected with 500 µg/ml G418 sulfate (Sigma-Aldrich, A1720). For inducible expression of PAR-T, the cells were treated with 1 µg/ml doxycycline (Dox) for 24 hr.

## Preparation of cell lysates

Cells were cultured and treated as described above for the preparation of cell extracts. At the conclusion of the treatments, the cells were washed two times with ice-cold PBS and lysed with Lysis Buffer (20 mM Tris-HCl pH 7.5, 150 mM NaCl, 1 mM EDTA, 1 mM EGTA, 1% NP-40, 1% sodium deoxycholate, and 0.1% SDS) containing 1 mM DTT, 250 nM ADP-HPD (Sigma-Aldrich, A0627), 10 µM PJ34 (Enzo, ALX-270), and 1× complete protease inhibitor cocktail (Roche, 11697498001). The cells were incubated in the Lysis Buffer for 30 min on ice with gentle vortexing and then centrifuged at full speed for 15 min at 4°C in a microcentrifuge to remove the cell debris.

## Western blotting

Protein concentrations of the cell lysates were determined using a Bio-Rad Protein Assay Dye Reagent (Bio-Rad, 5000006). Volumes of lysates containing equal amounts of total protein were boiled at 100°C for 5 min after the addition of 1/4 volume of 4× SDS-PAGE Loading Solution (250 mM Tris, pH 6.8, 40% glycerol, 0.04% Bromophenol Blue, and 4% SDS), run on 6% polyacrylamide-SDS gels, and transferred to nitrocellulose membranes. After blocking with 5% nonfat milk in TBST, the membranes were incubated with the primary antibodies described below in 1% nonfat milk in TBST with 0.02% sodium azide, followed by anti-rabbit HRP-conjugated IgG (1:5000) or anti-mouse HRP-conjugated IgG (1:5000). Immunoblot signals were detected using an ECL detection reagent (Thermo Fisher Scientific, 34577, 34095).

## Antibodies

The custom rabbit polyclonal antiserum against PARP-1 was generated in-house by using purified recombinant amino-terminal half of PARP-1 as an antigen (now available Active Motif; cat. no. 39559).

The custom recombinant antibody-like anti-poly-ADP-ribose binding reagent (anti-PAR) was generated and purified in-house (now available from EMD Millipore, MABE1031). The other antibodies used were as follows: PARP-2 (Santa Cruz Biotechnology, sc-150X), β-Tubulin (Abcam, ab6046), and goat anti-rabbit HRP-conjugated IgG (Pierce, 31460).

## Purification of PAR-T sensor proteins expressed in bacteria

His-tagged PAR detection reagents in the pET19b-based bacterial expression vector were expressed in *Escherichia coli* strain BL21(DE3-pLysis). The transformed bacteria were grown in LB containing ampicillin at 37°C until the $OD_{595}$ reached 0.4–0.6. Recombinant protein expression was induced by the addition of 1 mM IPTG for 3 hr at 37°C. The cells were collected by centrifugation, and the cell pellets were flash-frozen in liquid nitrogen and stored at –80°C until further use. The frozen cell pellets were thawed on ice and lysed by sonication in Ni-NTA Lysis Buffer (10 mM Tris-HCl pH 7.5, 0.15 M NaCl, 0.5 mM EDTA, 0.1% NP-40, 10% glycerol, 1 mM PMSF, and 1 mM β-mercaptoethanol). The lysates were clarified by centrifugation at 15,000 rpm using an SS34 rotor (Sorvall) at 4°C for 30 min. The supernatant was incubated with 1 ml of Ni-NTA resin equilibrated in Ni-NTA Lysis Buffer at 4°C for 2 hr with gentle mixing. The resin was collected by centrifugation at 4°C for 10 min at 1000×*g*, and the supernatant was removed. The resin was washed five times with Ni-NTA Wash Buffer (10 mM Tris-HCl pH 7.5, 0.3 M NaCl, 0.2% NP-40, 10% glycerol, 15 mM imidazole, 1 mM PMSF, and 1 mM β-mercaptoethanol). The recombinant proteins were then eluted using Ni-NTA Elution Buffer (10 mM Tris-HCl pH 7.5, 0.2 M NaCl, 0.1% NP-40, 10% glycerol, 500 mM imidazole, 1 mM PMSF, and 1 mM β-mercaptoethanol). The eluates were collected by centrifugation at 4°C for 10 mins at 1000×*g*, and dialyzed in Ni-NTA Dialysis Buffer (10 mM Tris-HCl pH 7.5, 0.15 M NaCl, 10% glycerol, 1 mM PMSF, and 1 mM β-mercaptoethanol). The dialyzed proteins were quantified using a Bradford protein assay (Bio-Rad), aliquoted, flash-frozen in liquid $N_2$, and stored at −80°C.

## Detection of PAR by PAR-T sensors in vitro

The following methods were used to measure PAR levels using the recombinant PAR-T constructs in vitro.

### In vitro auto ADP-ribosylation assays

In vitro auto ADP-ribosylation assays were performed essentially as described previously (*Gibson et al., 2017*). The ADP-ribosylation reactions contained 0.2 μM purified PARP-1 or PARP-3 and 100 ng/μl of sheared salmon sperm DNA (Invitrogen, AM9680). Purified proteins were mixed with the indicated amounts of $NAD^+$ in ADP-ribosylation Buffer (50 mM Tris-HCl pH 7.5, 0.125 M NaCl, and 12.5 mM $MgCl_2$). The reactions were incubated at room temperature for 15 min and terminated by addition of 200 nM PJ34.

### Tracking PAR formation in vitro

For characterizing the specificity of PAR tracking proteins, recombinant PAR-T sensor proteins (200 nM) were incubated with either of the following: 500 μM $NAD^+$, 500 μM NAM, 50 μM free ADP-ribose, 500 μM NMN, 5 mM ATP, and 100 μM NADH ATP, or with 5 nM PARP-1 or PARP-3 proteins in the ADP-ribosylation Buffer (50 mM Tris-HCl pH 7.5, 0.125 M NaCl, and 12.5 mM $MgCl_2$) for 15 min at room temperature followed by spectroscopy.

For time course experiments, in vitro ADP-Ribosylation assays were performed as described above in the presence of 200 nM of recombinant PAR-T sensor proteins. About 250 μM $NAD^+$ was added and the reaction was allowed to proceed for 16 min. The fluorescence intensities were measured every 30 s using a plate reader (CLARIOstar BMG Labtech). A similar in vitro ADP-ribosylation reaction was performed and processed for Western blotting as described above.

### Tracking PAR degradation in vitro

For measuring the level of PAR degradation in vitro, 5 nM of PARP-1 and 200 nM recombinant PAR-T sensor proteins were incubated in ADP-ribosylation Buffer along with the indicated amounts of ARH3. The reaction mixture was incubated at 37°C for the indicated amount of time and fluorescence intensities were measured.

### Tracking PAR formation in cell extracts

For measuring the levels of PAR in mammalian cell lysates, HeLa cells subjected to Dox-induced expression of the PAR-T sensor proteins were treated with 20 µM PJ34 or 20 µM PDD 00017273 for 2 hr prior to induction of DNA damage by treatment with $H_2O_2$ (1 mM; Sigma-Aldrich, 216763) for 10 min. The cells were then lysed in Lysis Buffer (20 mM Tris-HCl pH 7.5, 150 mM NaCl, 1 mM EDTA, 1 mM EGTA, 1% NP-40, 1% sodium deoxycholate, and 0.1% SDS) containing 1 mM DTT, 250 nM ADP-HPD (Sigma-Aldrich, A0627), 10 µM PJ34 (Enzo, ALX-270), and 1× complete protease inhibitor cocktail (Roche, 11697498001). Equal volumes of the lysate and ADP-ribosylation Buffer were incubated with 200 nM of recombinant PAR sensor proteins. The reaction mixture was incubated at room temperature for 15 min and the fluorescence intensity was measured.

### Measuring the levels of immobilized PAR in a plate-based assay

Plate-based fluorescent assays using PAR-T ddGFP were performed in a manner similar to the ELISA assays according to the manufacturer's protocol (Cell Biolabs, XDN-5114). Briefly, the protein binding strip-wells were coated with the anti-PAR Coating Antibody (1:500 in phosphate-buffered saline [PBS]) overnight. The antibody was removed and the wells were blocked with the Assay Diluent for 2 hr at room temperature. The solution was removed and the wells were washed three times with PBS. The wells were then incubated with the indicated amounts of PAR in Assay Diluent for 1 hr at room temperature with gentle shaking. The wells were then washed three times with 1× wash buffer and incubated with 200 nM sensor protein in ADP-ribosylation Buffer (50 mM Tris-HCl pH 7.5, 0.125 M NaCl, and 12.5 mM $MgCl_2$) for 30 min at room temperature in the dark and the fluorescence intensity was measured. The intensity from a blank well was used to determine background signal, which was subtracted from the fluorescence signal.

### Measurement of in vitro fluorescence changes using spectroscopy

Purified sensor and control proteins (200 nM) were incubated with the purified proteins or cell extracts in a volume of 100 µl. The samples were incubated for 15 min and the fluorescence intensity was measured using a plate reader (CLARIOstar BMG Labtech). Excitation and emission spectra were 488 and 530 nm, respectively.

## Immunofluorescence and live-cell imaging assays

The following methods were used for measuring the PAR levels in live cells using confocal microscopy.

## Tracking PAR in live cells

For live-cell imaging assays to measure the level of PAR, HeLa cells expressing the fluorescence PAR-T sensors were plated on poly-D-Lysine coated live-cell imaging chamber slides (Thermo Fisher Scientific, 15411) and cultured in FluoroBrite medium (Thermo Fisher Scientific, A1896701) supplemented with 10% FBS (TET tested; Atlanta Biologicals, S103050) and 1% penicillin/streptomycin and 1 mg/mL of Dox. After 24 hr, the cells were pre-treated with 20 µM PJ34 for 2 hr followed by treatment with 1 mM $H_2O_2$ for the indicated times. Images were acquired using an inverted Zeiss LSM 780 confocal microscope affixed with a 37°C, 5% $CO_2$ incubator.

### Image analysis

We used ImageJ software to subtract background, set thresholds, select the regions of interest (ROIs), and quantify fluorescence intensity. The nuclei were selected using mCherry signal and corresponding intensity of PAR (GFP signal) was quantified. Time series v.3.0 plugin was used to get an average of the ROI intensities of mCherry and GFP.

## Generation of 3D cancer spheroids

For generation of cancer spheroids, MCF-7 cells expressing the fluorescence PAR-T sensors were sorted by fluorescence-activated cell sorting to obtain a population of cells expressing high levels of mCherry (a marker for nuclei). The 3D cancer spheroids were then generated in Matrigel as described previously (*Debnath et al., 2003*) with some variations. The bottom of an eight-well chamber slide (Thermo Fisher Scientific, 154534) was coated with 90 µl of 100% growth factor reduced Matrigel (Thermo Fisher Scientific, CB 40230) and incubated at 37°C for 30 min to allow the Matrigel to solidify.

Then 5000 cells in 400 µl of complete Fluorobrite medium containing 2% Matrigel were added on top of the Matrigel. The medium was changed every 3 days and the cells were allowed to grow for 2 weeks. Once the cells formed 3D clusters, expression of the PAR-T sensor was induced by treating the cells with 1 mg/ml Dox for 24 hr. For inhibition of PARylation, the cells were treated with 20 µM Niraparib for 24 hr. For the time course experiments, the cancer cell spheroids were generated as described above in a 96-well plate format (Cellvis, P96-1-N). The spheroids were then treated with 20 µM Niraparib and imaged at the indicated times. Images of the 3D spheroids were acquired using an LSM 880 confocal microscope.

### Image analysis

Multiple z-stacks were acquired and the z-stack projections were obtained using Fiji ImageJ software. The spheroids were selected using the mCherry signal and the intensity of the corresponding PAR (GFP signal) was quantified. To obtain the intensities of the signals in the outer layer and core of the spheroids, the intensities of whole spheroid and the cores were measured and the following formula was used to calculate the PAR signal from cells within the outer layer:

$$\frac{(\text{GFP fluorescence of whole spheroid} * \text{Area of whole spheroid}) - (\textit{GFP fluorescence of outer layer} * \textit{Area of outer layer})}{(\text{mCherry fluorescence of whole spheroid} * \text{Area of whole spheroid}) - (\textit{mCherry fluorescence of outer layer} * \textit{Area of outer layer})}$$

## Immunofluorescent staining of cultured cells

293T cells were seeded into eight-well chambered slides (Thermo Fisher Scientific, 154534). After 24 hr, the cells were treated with 20 µM PJ34 for 2 hr followed by $H_2O_2$ treatment for 15 min. The cells were then washed once with PBS, fixed with 4% paraformaldehyde for 15 min at room temperature, and washed three times with PBS. The cells were permeabilized for 5 min using Permeabilization Buffer (PBS containing 0.1% Triton X-100), washed three times with PBS, and incubated for 1 hr at room temperature in Blocking Solution (PBS containing 1% BSA, 10% FBS, 0.3 M glycine, and 0.1% Tween-20). The fixed cells were incubated with PAR antibody (1:500 Millipore, MABE1031) in PBS overnight at 4°C. After washing three times with PBS, the cells were incubated with Alexa Fluor 594 goat anti-rabbit IgG (1:500, Thermo Fisher Scientific, A-21207) in PBS for 1 hr at room temperature. After washing three times with PBS, coverslips were mounted with the VectaShield Antifade Mounting Medium with DAPI (Vector Laboratories, H-1200). All images were acquired using an inverted Zeiss LSM 880 confocal microscope. Fiji ImageJ software was used to perform the quantification of fluorescence intensities. The ROUT method in GraphPad PRISM was used to identify and remove the outliers.

## Luciferase assays

Luciferase assays in live cells were performed to measure the levels of PAR in cells. Cells subjected to dox-inducible PAR-T expression were plated into black bottomed 96-well plates in the presence of 1 mg/ml Dox. After 24 hr, the cells were treated as indicated and luciferase assays were performed using the Nano-Glo Live Cell Assay System (Promega, N2011) for Nanoluciferase measurements or by using the Luciferase assay system (Promega, E1500) for measuring firefly luciferase measurements according to the manufacturer's instructions. For dual-luciferase assays, images of PAR-T NanoLuc were first obtained using the Nano-Glo Live Cell Assay System. The medium was then changed to PBS containing 100 µg/ml of D-luciferin (Gold Biotechnology, LUCNA-500) to obtain luminescence images of firefly luciferase. The luminescence images were obtained using IVIS Spectrum with an open filter. The luminescence intensities were quantified either by measuring the ROI intensities using IVIS Spectrum or by spectroscopy using a plate reader (CLARIOstar BMG Labtech).

## Time course experiments

To detect the time-dependent changes in PARP-1 activation upon UV treatment, MDA-MB-231 cells were plated into black bottomed 96-well plates in the presence of 1 mg/ml Dox. After 24 hr, the cells were treated with 20 µM Niraparib or 20 µM PARG inhibitor in FluoroBrite medium (Thermo Fisher Scientific) supplemented with 10% FBS for 2 hr. The cells were then treated with UV radiation and then luciferase assays were performed using the Nano-Glo Live Cell Assay System and the luminescence intensities were measured in an interval of 1 min using the CLARIOstar plate reader. The data was shown as a mean of the ratio of luminescence from PAR-T NanoLuc to that of intact nano luciferase.

### Enzyme-linked immunosorbent assays

ELISAs were performed according to the manufacturer's protocol (Cell Biolabs, XDN-5114). Briefly, the protein binding strip-wells were coated with the anti-PAR Coating Antibody (1:500 in PBS) overnight. The antibody was removed and the wells were blocked with the Assay Diluent for 2 hr at room temperature. The solution was removed and the wells were washed three times with PBS. The wells were then incubated with the indicated amounts of PAR in the Assay Diluent for 1 hr at room temperature with gentle shaking. The wells were then washed three times with 1× wash buffer and incubated with anti-PAR Detection Antibody (1:1000 in Assay Diluent) for 1 hr at room temperature with gentle shaking. The wells were washed three times with 1× wash buffer and incubated with the Secondary Antibody-HRP conjugate (1:1000 in Assay Diluent) for 1 hr at room temperature with gentle shaking. The wells were washed three times with 1× wash buffer, incubated with the substrate solution. The reaction was stopped with Stop Solution when the color change was observed and the absorbance at 450 nm was measured.

### Xenograft experiments

All mouse-based experiments were performed in compliance with the Institutional Animal Care and Use Committee (IACUC; protocol no. 2015-101155) at the UT Southwestern Medical Center.

### Assays with established xenografts

To establish breast cancer xenografts, $10 \times 10^6$ MDA-MB-231-Luc cells with Dox-inducible expression of PAR-T NanoLuc were injected subcutaneously into the flank of female NOD/SCID/gamma (NSG) mice at 6–8 weeks of age in 100 µl of 1:1 ratio of PBS and Matrigel (Thermo Fisher Scientific, CB 40230). The weight of the mice and tumor growth was monitored once per week. Once the tumors were formed (~3 weeks post-tumor cell injection), the mice were randomized to receive 25 mg/kg of Niraparib or PARG inhibitor or vehicle in 4% DMSO, 5% PEG 300, 5% Tween80 in PBS intraperitoneally for 5 consecutive days a week (2 days off) for 3 weeks. The mice were placed on a Dox containing diet (625 mg/kg; Envigo) and injected with 100 µg/g of Dox for 5 days a week, for 3 weeks. Bioluminescence imaging of the xenografts was performed using IVIS Lumina. The mice were anesthetized, treated locally with 5 Gy of radiation, followed by injection in PBS of a combination of 40× dilutions of Nano-Glo live-cell imaging (Promega, N2011) and Nano-Glo (Promega, N1110) substrates for acquiring Nanoluciferase luminescence. Luminescence from firefly luciferase was then measured by treating the mice with 150 mg/kg D-Luciferin.

### Time course assay

For the time course experiment ± PARG inhibitor, $10 \times 10^6$ MDA-MB-231-Luc cells with Dox-inducible expression of PAR-T NanoLuc were injected subcutaneously into the flank of the C57/BL6 mice in 100 µl of 1:1 ratio of PBS and Matrigel. The mice were immediately treated with 25 mg/kg of PARG inhibitor or vehicle in 4% DMSO, 5% PEG 300, 5% Tween80 in PBS. Bioluminescence imaging of the xenografts was performed using IVIS Spectrum. The mice were anesthetized, followed by injection in PBS of a combination of 40× dilutions of Nano-Glo live-cell imaging substrate and Nano-Glo substrate for acquiring Nano luciferase luminescence 6 or 24 hr after the injection of the cells.

## Acknowledgements

The authors thank Dr. Rebecca Gupte for technical assistance in purifying the recombinant proteins and for critical comments on this manuscript. The authors acknowledge and thank the following UT Southwestern core facilities: Live-Cell Imaging Core for microscopy support (Dr. Katherine Luby-Phelps) and Flow Cytometry Core for performing FACS (Dr. David Farrar). The authors would like to acknowledge the assistance of the Southwestern Small Animal Imaging Shared Resource, which is supported in part by the Harold C. Simmons Cancer Center through an NCI Cancer Center Support Grant, P30 CA142543. This work was supported by a grant from the NIH/National Institute of Diabetes and Digestive and Kidney Diseases (NIDDK) (R01 DK058110), a grant from the Cancer Prevention and Research Institute of Texas (RP190236), and funds from the Cecil H. and Ida Green Center for Reproductive Biology Sciences Endowment to WLK.

## Additional information

### Competing interests

Sridevi Challa, Keun W Ryu: K.W.R., S.C., and W.L.K. have a patent pending for the PAR-T sensors described herein. W Lee Kraus: W.L.K. is a founder and consultant for Ribon Therapeutics, Inc and ARase Therapeutics, Inc He is also coholder of U.S. Patent 9,599,606 covering the ADP-ribose detection reagent used herein, which has been licensed to and is sold by EMD Millipore. The other authors declare that no competing interests exist.

### Funding

| Funder | Grant reference number | Author |
|---|---|---|
| National Institute of Diabetes and Digestive and Kidney Diseases | R01 DK058110 | W Lee Kraus |
| Cancer Prevention and Research Institute of Texas | RP190236 | W Lee Kraus |

The funders had no role in study design, data collection and interpretation, or the decision to submit the work for publication.

### Author contributions

Sridevi Challa, Conceptualization, Data curation, Formal analysis, Investigation, Methodology, Validation, Visualization, Writing – original draft, Writing – review and editing; Keun W Ryu, Conceptualization, Data curation, Formal analysis, Investigation, Methodology, Validation, Visualization, Writing – review and editing; Amy L Whitaker, Data curation, Formal analysis, Investigation; Jonathan C Abshier, Investigation; Cristel V Camacho, Data curation, Formal analysis, Investigation, Methodology; W Lee Kraus, Conceptualization, Funding acquisition, Project administration, Supervision, Writing – review and editing

### Author ORCIDs

Cristel V Camacho ![iD] http://orcid.org/0000-0003-1723-579X
W Lee Kraus ![iD] http://orcid.org/0000-0002-8786-2986

### Ethics

All mouse xenograft experiments were performed in compliance with the Institutional Animal Care and Use Committee (IACUC; protocol no. 2015-101155) at the UT Southwestern Medical Center.

### Decision letter and Author response

Decision letter https://doi.org/10.7554/eLife.72464.sa1
Author response https://doi.org/10.7554/eLife.72464.sa2

---

## Additional files

### Supplementary files

• Transparent reporting form
• Supplementary file 1.

### Data availability

All unprocessed Western blot image data and data from individual replicates from the fluorescent and luminescent assays can be found on Mendeley at https://data.mendeley.com/datasets/x9j73tdb5r/1. The work does not contain any high complexity, high content "omics" data.

The following dataset was generated:

| Author(s) | Year | Dataset title | Dataset URL | Database and Identifier |
|-----------|------|---------------|-------------|-------------------------|
| Kraus WL, Challa S, Ryu KW | 2022 | Development and Characterization of PAR-Trackers: New Tools for Detecting Poly(ADP-ribose) In Vitro and In Vivo | https://data.mendeley.com/datasets/x9j73tdb5r/1 | Mendeley Data, 10.17632/x9j73tdb5r.1 |

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
