## [Editor Report]

Challa and Ryu et al., systematically evaluated various combinations of ADP-ribose-binding modules to make sensors detecting poly(ADP-ribose). A series of GFP- and luciferase-based sensors has been created by the authors who demonstrated their applications in vitro, in living cells, and, for the first time, in animals. Despite these sensors still needing to be improved upon their dynamic ranges and signal/background ratios, these tools open the possibility to detect ADP-ribosylation in many experimental and biological systems in vitro and in vivo.

---

## [Decision Letter]

**Decision letter after peer review:**

Thank you for submitting your article "Development and Characterization of PAR-Trackers: New Tools for Detecting Poly(ADP-ribose)" for consideration by *eLife*. Your article has been reviewed by 3 peer reviewers, including Anthony K. L. Leung as Reviewing Editor and Reviewer #1, and the evaluation has been overseen by Philip Cole as the Senior Editor.

Essential revisions:

1) The manuscript will be improved with quantitative comparisons with existing techniques, such as western blot, immunofluorescence, and ELISA.

2) The manuscript will be improved to clarify the dynamic range for both sensors and the high background observed in the luciferase construct.

3) "We demonstrate that these new tools allow the detection and quantification of PAR levels in extracts, living cells, and living tissues with greater sensitivity, as well as temporal and spatial precision." The claim for their PAR sensors in cells is robust, but the current form of the manuscript has yet to demonstrate the spatio-temporal dynamics in animals. Time-course experiments and multiple spheroids of cancer cells to indicate the ability to detect the temporal and spatial dynamics will support that claim.

4) The manuscript will be improved with better descriptions of methods and statistical analyses.

*Reviewer #1 (Recommendations for the authors):*

The manuscript may be improved by adding a discussion on how to improve their tools to increase their dynamic range, sensitivity and background.

*Reviewer #2 (Recommendations for the authors):*

1) A more direct, quantitative analysis would be important. For example, the ddGFP assays (Figures 1E/G) are quantitative but the blots (Figure 1D/F) are not. The authors should compare the ddGFP assay to a PAR ELISA, for example, so as to directly compare more effectively. This would be helpful in understanding why the value in Figure 1G does not reach to baseline.

2) Figure 2 – Authors should indicate the time and dose of hydrogen peroxide.

3) Figure 2C/D – the n values indicated – is that the number of cells? How many different experiments total? The standard would be >30 cells from at least two separate experiments.

4) Figure 2 – supplement. The spheroid experiment is interesting but only one spheroid is shown and in fact with PARPi treatment the signal is not completely blocked. Would be ideal to develop this further for more quantitative analysis.

5) Figure 3C and F – the high level of signal absent a stimulus is surprising. Can the authors explain this? It would appear to be a high level of background or non-specific activity.

*Reviewer #3 (Recommendations for the authors):*

1. I recommend exploring the capability of PAR-T GFP to detect the spatiotemporal dynamics of the PAR process in cancer spheroid in different conditions.

2. The split nanoluc technology may be used to detect the dynamics of the targeted analyte. Thus the authors might want to perform time-course imaging of PAR levels in mice in response to different drug treatments using PAR-T Luc.

---

## [Author Response]

Essential revisions:1) The manuscript will be improved with quantitative comparisons with existing techniques, such as Western blot, immunofluorescence, and ELISA.

We agree with the reviewers that a comparison with existing PAR detection technologies will improve the manuscript. We have now performed a comprehensive set of assays to compare the performance of the PAR-T sensors to established PAR detection methodologies. In sum, we compared:

1. Western blotting with WWE-Fc versus fluorescence assay with PAR-T ddGFP, which were performed in conjunction with ARH3-mediated degradation of PAR in vitro (Figure 6A).

2. Western blotting with WWE-Fc versus live cell luciferase assay using PAR-T NanoLuc, which were performed in conjunction with UV-induced DNA damage in MDA-MB-231 cells (Figure 6B).

3. ELISA with PAR antibody versus fluorescence assay with PAR-T ddGFP, which was performed using immobilized PAR (Figure 6C).

4. Immunofluorescence with WWE-Fc versus live cell imaging using PAR-T ddGFP, which was performed using H2O2-mediated PARP1 activation (Figure 6D).

The results from these assays demonstrate that the PAR-T sensors function comparably to the other available PAR detection techniques. We included an in-depth discussion of these results in our response to the editor’s second comment.

2) The manuscript will be improved to clarify the dynamic range for both sensors and the high background observed in the luciferase construct.

We noted the reviewers’ concerns about the high background signal in Niraparib-treated samples. To address this issue, we compared the dynamic range of PAR-T NanoLuc with Western blotting (Figure 6B, Figure 5A-5C) and observed that the results from PAR-T NanoLuc are comparable to Western blotting. Of note, we were able to detect a decrease in PAR levels with Niraparib in the absence of DNA damage using PAR-T NanoLuc, but not Western blotting. Based on these analyses, we can conclude that the changes in PAR levels at the basal level are very minimal, leading to only 50% decrease in PAR-T NanoLuc signal with Niraparib treatment. Note that the decrease in PAR-T NanoLuc signal is greater when UV-treated cells were pre-treated with Niraparib (Figure 5A).

We have now performed a set of assays to compare the performance of the PAR-T sensors to conventional PAR detection reagents (WWE-Fc and PAR antibody) in a variety of assays (Figure 6E). A comparison of the techniques available to detect PAR showed that Western blotting with WWE-Fc has the highest dynamic range for detection of PAR (8-fold), but the dynamic range of live cell luciferase assay with PAR-T NanoLuc is comparable to that of Western blotting (6-fold). While PAR-T ddGFP in a modified fluorescence assay has a better dynamic range than PAR antibody in an ELISA (6-fold vs 3.5-fold) when the assays were performed using immobilized PAR, it has a lower dynamic range when used for live cell imaging (4.4-fold). This can be explained in-part due to the higher auto-fluorescence of cells that can diminish the dynamic of the PAR-T ddGFP sensors. We understand that this is a limitation of the sensor, but it is beyond the scope of the current manuscript to improve it further. Here, we focus on characterizing these sensors for the first time and look forward to improving them in future studies. Nevertheless, the performance of PAR-T ddGFP in live cells is comparable to that of an immunofluorescence assay with WWE-Fc (4.4-fold vs 5-fold).

3) "We demonstrate that these new tools allow the detection and quantification of PAR levels in extracts, living cells, and living tissues with greater sensitivity, as well as temporal and spatial precision." The claim for their PAR sensors in cells is robust, but the current form of the manuscript has yet to demonstrate the spatio-temporal dynamics in animals. Time-course experiments and multiple spheroids of cancer cells to indicate the ability to detect the temporal and spatial dynamics will support that claim.

We agree with the reviewer’s comment that the original manuscript did not demonstrate that the PAR-T sensors can be used to detect spatio-temporal changes in PAR. To address this, we included a video from live cell imaging showing a time-dependent, H2O2-mediated increase in PAR-T ddGFP (Figure 2 —figure supplement 2, video). We also performed a time course in 3D cancer spheroids to visualize the spatio-temporal changes in PAR levels in response to Niraparib (Figure 3C and 3D). The results from this experiment show that the PAR levels in cells at the core of the spheroids are relatively resistant to Niraparib treatment, since the PAR levels in these cells decrease at a lower rate compared to PAR in the cells at the outer layer of the spheroid.

To demonstrate that PAR-T NanoLuc can be used to detect time-dependent changes in PAR levels, we performed a time course of UV-mediated PARP-1 activation in vitro (Figure 5D) and PARGi-mediated PAR accumulation in vivo (Figure 8 —figure supplements 1B-1D). The time course of UV-induced PARP-1 activation in cells demonstrated that the dynamic changes in PAR in response to DNA damage in living cells can be captured using the PAR-T NanoLuc sensors. In the in vivo studies with PAR-T NanoLuc, we observed an increase in PAR levels in response to PARG inhibition by 6 hours, with the signal diminishing within 24 hours. These results demonstrate that the PAR-T NanoLuc sensors can be used for detecting temporal changes in PAR in vivo.

4) The manuscript will be improved with better descriptions of methods and statistical analyses.

We provided more details for the description of methods and statistical analysis.

Reviewer #1 (Recommendations for the authors):The manuscript may be improved by adding a discussion on how to improve their tools to increase their dynamic range, sensitivity and background.

We included the following discussion on how to improve these tools:

“Although the dynamic range of the PAR-T sensors are comparable to other available PAR detection tools, the sensors can be improved further by optimizing the assay conditions and the design of the PAR-T sensors. For example, our data suggests that in plate-based fluorescence assays, immobilizing PAR on the well increases the sensitivity of detection by PAR-T ddGFP. The reduced dynamic range of PAR-T ddGFP in live cell imaging assays can be attributed to high background autofluorescence from the cells and culture medium. To minimize this background signal, future PAR-T fluorescent sensors should be developed using fluorophores that have low background signal, such as those that emit fluorescence at a far-red wavelength. In addition, the ddGFP-A fragment has low fluorescence intensity, possibly contributing to reduced dynamic range. To avoid this, an optimized split-fluorescence sensor should be used. Similarly, the background signals contributing to reduced dynamic range in the assays using the PAR-T NanoLuc sensor can be reduced by optimizing the luciferase-fluorophore reporter pair used in the sensor.”

Reviewer #2 (Recommendations for the authors):1) A more direct, quantitative analysis would be important. For example, the ddGFP assays (Figures 1E/G) are quantitative but the blots (Figure 1D/F) are not. The authors should compare the ddGFP assay to a PAR ELISA, for example, so as to directly compare more effectively. This would be helpful in understanding why the value in Figure 1G does not reach to baseline.

We agree with the reviewer that a comparison with existing PAR detection technologies will improve the manuscript. We now performed a comparison of ELISA, Western blotting, and immunofluorescence assays with the fluorescence assays and live cell imaging using PAR-T ddGFP (Figure 6). The results indicate that the detection range of PAR-T ddGFP based assays are comparable to the established PAR detection assays.

2) Figure 2 – Authors should indicate the time and dose of hydrogen peroxide.

We added this information.

3) Figure 2C/D – the n values indicated – is that the number of cells? How many different experiments total? The standard would be >30 cells from at least two separate experiments.

We previously indicated that they were from 3 biological replicates. We now corrected the text to show that the bar graphs represent mean fluorescence of > 50 cells in Figure 2B, and, 20 cells in Figure 2D.

4) Figure 2 – supplement. The spheroid experiment is interesting but only one spheroid is shown and in fact with PARPi treatment the signal is not completely blocked. Would be ideal to develop this further for more quantitative analysis.

We noted the reviewer’s concern and performed quantitative analysis of multiple spheroids. As shown in Figure 3B, we observed a significantly higher GFP fluorescence signal in spheroids derived from PAR-T ddGFP expressing cells compared to those expressing ddGFP or the spheroids treated with Niraparib.

We also developed an analysis approach that allows us to quantify the signals from the core of the spheroids separately from the periphery of the spheroids. We performed a time course in 3D cancer spheroids to visualize the spatio-temporal changes in PAR levels (Figure 3C and 3D). The results from this experiment demonstrate that the PAR levels in cells at the core of the spheroids are relatively resistant to Niraparib treatment, as the PAR levels in cells at the core of the spheroid decrease at a lower rate when compared to PAR in the cells at the outer layer of the spheroid.

5) Figure 3C and F – the high level of signal absent a stimulus is surprising. Can the authors explain this? It would appear to be a high level of background or non-specific activity.

We appreciate the reviewer’s concern about the high background signal in Niraparib treated samples. To answer this concern, we compared the dynamic range of Western blotting using WWE-Fc and live cell luciferase assay using PAR-T NanoLuc (Figure 6B). We observed that the results from PAR-T NanoLuc based luciferase assay are comparable to Western blotting. Of note, we were able to detect decrease in PAR levels with Niraparib using PAR-T NanoLuc based luciferase assay but not Western blot analysis. Based on these analyses, we can conclude that the changes in PAR levels at the basal level are very minimal, leading to only 50% decrease in PAR-T NanoLuc signal with Niraparib treatment (Figure 6B, Figure 5A-5C). Note that the decrease in live cell luciferase assay using PAR-T NanoLuc signal is greater when UV-treated cells were pre-treated with Niraparib, which is consistent with the results from Western blot analysis (Figure 5A).

Reviewer #3 (Recommendations for the authors):1. I recommend exploring the capability of PAR-T GFP to detect the spatiotemporal dynamics of the PAR process in cancer spheroid in different conditions.

See our response to Comment 2 above. To address the reviewer’s concern about using PAR-T ddGFP for spatio-temporal changes in cells, we included a video for live cell imaging of H2O2-mediated increase in PAR-T ddGFP (Figure 2 —figure supplement 2, video).

We also performed a time course in 3D cancer spheroids to visualize the spatio-temporal changes in PAR levels (Figure 3C and 3D). The results from this experiment demonstrate that the PAR levels in cells at the core of the spheroids are relatively resistant to Niraparib treatment, as the PAR levels in cells at the core of the spheroid decrease at a lower rate when compared to PAR in the cells at the outer layer of the spheroid.

2. The split nanoluc technology may be used to detect the dynamics of the targeted analyte. Thus the authors might want to perform time-course imaging of PAR levels in mice in response to different drug treatments using PAR-T Luc.

To demonstrate that PAR-T NanoLuc can be used to detect time-dependent changes in PAR levels, we performed a time course of UV-mediated PARP-1 activation (Figure 5D). The results from this assay demonstrated that the dynamic changes in PAR in live cells in response to DNA damage can be captured using the PAR-T NanoLuc sensors.

In addition, we also measured PARGi-mediated PAR accumulation in vivo in xenograft tumors (Figure 8 —figure supplement 1B-1D). We found that PAR can be detected readily in breast cancer cells when injected into mice. Upon treatment with PARGi, the luminescence from PAR-T NanoLuc increased significantly by 6 hours and then diminished by 24 hours. These data demonstrate that PAR-T NanoLuc can be used to track dynamic changes in PAR levels both in cells and in animals.